# CateKV: On Sequential Consistency for Long-Context LLM Inference Acceleration

Haoyun Jiang [1 2]   Haolin Li [3]   Jianwei Zhang [2]   Fei Huang [2]   Qiang Hu [1]   Minmin Sun [2]   Shuai Xiao [2]   Yong Li [2]
Junyang Lin [2]   Jiangchao Yao [1]

## Abstract

Large language models (LLMs) have demonstrated strong capabilities in handling long-context tasks, but processing such long contexts remains challenging due to the substantial memory requirements and inference latency. In this work, we discover that certain attention heads exhibit sequential consistency in their attention patterns, which can be persistently identified using a coefficient-of-variation-based algorithm. Inspired by this observation, we propose CateKV, a hybrid KV cache method that retains only critical token information for consistent heads, thereby reducing KV cache size and computational overhead, while preserving the majority of KV pairs in adaptive heads to ensure high accuracy. We show the unique characteristics of our algorithm and its extension with existing acceleration methods. Comprehensive evaluations on long-context benchmarks show that, while maintaining accuracy comparable to full attention, CateKV reduces memory usage by up to $2.72\times$ and accelerates decoding by $2.18\times$ in single-sample inputs, and boosts throughput by $3.96\times$ in batch scenarios.

## 1. Introduction

With rapid development of large language models (LLMs), many generalist models support context windows of 128K tokens or more (Achiam et al., 2023; Dubey et al., 2024; Bai et al., 2023a; GLM et al., 2024b; Team et al., 2023; Abdin et al., 2024), enabling to effectively perform tasks like long-document question answering (Caciularu et al., 2023; Wang et al., 2024), information retrieval (Zhang et al., 2024a; Zhu et al., 2023), and repository-level code under-

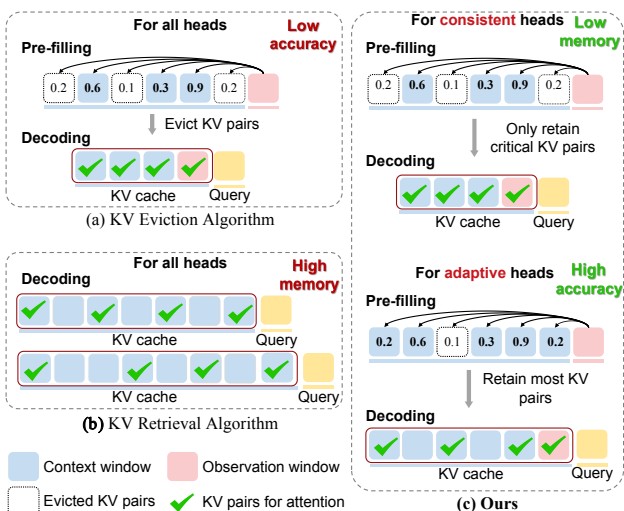

Figure 1: Comparison of KV eviction, KV retrieval, and our method. CateKV employs a hybrid KV cache method to reduce memory usage while maintaining high accuracy.

standing (Bairi et al., 2024; Jimenez et al., 2023). However, as context lengths grow, the autoregressive nature of the mainstream LLM paradigm often leads to increased computational costs, memory consumption, and thus the runtime, since we have to store and retrieve all key-value (KV) caches. For example, using the Llama-3-8B-Instruct-262k (Gradient, 2024b) model with FlashAttention, extending the context from 1K to 1M tokens increases inference latency by over 3000 times (Jiang et al., 2024). Therefore, accelerating LLM inference in long contexts is both essential and imperative.

Existing methods for inference acceleration of LLMs under long contexts can be categorized into two types: KV cache eviction and KV cache retrieval strategies. KV eviction strategies reduce the size of the KV cache by systematically discarding KV pairs based on predefined policies (Xiao et al., 2023; Zhang et al., 2023; Li et al., 2024; Cai et al., 2024). However, this usually suffers from significant accuracy loss, as the removal of essential information without comprehensive contextual understanding adversely affects task performance. In contrast, KV retrieval methods maintain accuracy by preserving all KV pairs in the cache and

[1] CMIC, Shanghai Jiao Tong University [2] Alibaba Group [3] Fudan University. Correspondence to: Jiangchao Yao <sunarker@sjtu.edu.cn>, Shuai Xiao <shuai.xsh@gmail.com>.

*Proceedings of the $42^{nd}$ International Conference on Machine Learning*, Vancouver, Canada. PMLR 267, 2025. Copyright 2025 by the author(s).

selectively retrieving relevant tokens during the decoding stage, thereby ensuring that no crucial information is omitted (Tang et al., 2024b; Sun et al., 2024). Nevertheless, KV retrieval does not mitigate high memory usage, limiting scalable batch sizes and thus the throughput under long contexts. Furthermore, certain approaches (Jiang et al., 2024; Xiao et al., 2024b; Tang et al., 2024a) recognize the heterogeneous sparsity patterns across different attention heads within the model and configure the heads with distinct sparsification strategies during inference. While they show initial promise in accuracy and efficiency, they do not fully utilize the interplay between the pre-filling and decoding phases to facilitate further optimization.

Different from previous methods, we explore leveraging the experience in pre-filling stage to promote the decoding stage. Specifically, our intuition is inspired by an interesting observation: certain attention heads exhibit sequentially consistent attention patterns, spanning across the pre-filling and decoding stages, while some attention heads exhibit rich activation dynamics during the whole process, as shown in Figure 2. Besides, this phenomenon, reflecting the special head working mechanism, frequently existed in different LLMs and their different layers. This indicates that if we can capture those modes at the pre-filling stage, we can leverage the sequential consistency to only maintain a subset of crucial tokens for certain heads, which achieves both a reduction of the KV cache and the speedup of attention computation during the decoding stage for LLM inference acceleration.

Based on the above analysis, we propose CateKV, a simple, effective, and plug-and-play method designed to enhance LLM inference efficiency by leveraging sequential consistency. CateKV uses an observation window during the pre-filling stage to identify critical tokens and employs a coefficient-of-variation-based score to classify attention heads into consistent and adaptive types, based on a reference dataset. In consistent heads, CateKV retains only a small proportion of critical KV pairs, while in adaptive heads, it retains most, selecting tokens based on their importance derived from attention weights in the pre-filling stage, as shown in Figure 1. By such an effective routing manner guided by sequential consistency, our method maintains the performance merit of LLM inference and simultaneously enjoys the acceleration gain. Our method is orthogonal to and combinable with many existing acceleration approaches, and we conducted extensive experiments on widely used benchmarks including RULER (Hsieh et al., 2024), Longbench (Bai et al., 2023b), and NIAH (Kamradt, 2024), using models such as LLaMA-3-8B-Instruct-1048K (Gradient, 2024a), GLM-4-9B-1M (GLM et al., 2024a), LLaMA-3.1-8B (Meta AI, 2024) and Yi-9B-200K (AI et al., 2024) to demonstrate the effectiveness. In a nutshell, our contributions are summarized as follows:

- We identify that general sequential consistency exhibits in certain attention heads and dynamic activation in others, dividing the attention heads into consistent and adaptive heads, which naturally constructs the basic for decoding acceleration with pre-filling experience.

- We propose CateKV, a hybrid KV cache acceleration algorithm leveraging sequential consistency. Employing a coefficient-of-variation-based score, CateKV can precisely classify attention heads into two categories, enabling efficient KV cache eviction while closely approximating the performance of full attention.

- Extensive evaluations on popular benchmarks demonstrate that CateKV reduces memory usage and decoding latency by $2.72\times$ and $2.18\times$ for single inputs, increases throughput by $3.96\times$ in batch scenarios, while maintaining performance comparable to full attention. Further acceleration can be achieved by integrating our plug-and-play method with other approaches.

## 2. Related Work

### 2.1. KV Cache Eviction Algorithm

To save the significant time and space overhead as the input length increases, various approaches explore evicting tokens to reduce both memory usage and computational cost. StreamingLLM (Xiao et al., 2023) introduces the phenomenon of "attention sink" and supports longer sequence by retaining only the KV pairs of attention sinks and recent tokens. H2O (Zhang et al., 2023) employs a low-cost eviction strategy to maintain a fixed-size KV cache containing heavy-hitters, based on the sum of historical attention scores. SnapKV (Li et al., 2024) uses the last tokens in the prompt during the prefilling stage to select critical tokens for generation in the decoding stage. PyramidKV (Cai et al., 2024), PyramidInfer (Yang et al., 2024), and LazyLLM (Fu et al., 2024a) leverage attention distribution across layers to dynamically adjust cache size, making cache allocation more efficient. Other methods like MagicPig (Chen et al., 2024), Q-Hitter (Zhang et al., 2024b), ALISA (Zhao et al., 2024) and , which combine KV cache eviction with quantization, hashing algorithms, or sparse windows, can also improve inference efficiency. However, these methods induce nonnegligible performance degradation, as they potentially evict certain tokens that are crucial for future generation.

### 2.2. KV Cache Retrieval Algorithm

Unlike KV cache eviction algorithms, KV cache retrieval algorithms retain a complete KV cache and dynamically retrieve important KV pairs to reduce inference latency. Following PageAttention (Kwon et al., 2023), Quest (Tang et al., 2024b) divides tokens into pages and devises an approximate attention score to retrieve the most relevant

pages for the current decoding steps. InfLLM (Xiao et al., 2024a) adopts a strategy similar to Quest, offloading most of the cache to the CPU to support longer prompts. ShadowKV (Sun et al., 2024) enhances the storage efficiency of Quest by storing only low-rank key caches and offloading value caches, allowing inference with large batch sizes and context lengths. Others like SparQ (Ribar et al., 2023), InfiniGen (Lee et al., 2024), and Loki (Singhania et al., 2024), accelerate the selection of top-$k$ critical tokens by reducing the dimension. KV cache retrieval methods maintain performance by preserving the entire KV cache but inevitably incur increased inference latency and storage costs.

### 2.3. Head-wise Attention Classification

Another line of work classifies attention heads into distinct sparse patterns. MInference (Jiang et al., 2024) divides the attention into A-shape, Vertical-Slash, and Block-Sparse patterns, achieving acceleration during the *pre-filling* stage. RazorAttention (Tang et al., 2024a) and DuoAttention (Xiao et al., 2024b) split heads into retrieval heads and streaming heads to determine whether to implement full attention or an attention mechanism similar to StreamingLLM (Xiao et al., 2023). Methods like AdaKV (Feng et al., 2024) and HeadKV (Fu et al., 2024b) achieve more fine-grained classification by allocating different budgets to each attention head. These methods focus on the features within individual heads or their variations, but overlook the patterns of attention heads across the prefilling and decoding stages.

## 3. Method

### 3.1. Sequential Consistency of Attention Heads

In this paragraph, we present an interesting observation about the attention patterns across the pre-filling and decoding stages. To illustrate this, we randomly selected a text segment from Wikitext (Merity et al., 2016) as input for the LongChat-7B (Li et al., 2023) model, examining how attention weights evolve throughout the generation process. Figure 2 illustrates the attention heatmaps during the pre-filling and decoding stages for two types of attention heads. We observe that for certain attention heads, attention is concentrated on a few critical tokens, which show clear consistency across both the pre-filling and decoding stages. This consistency allows us to identify critical tokens during the pre-filling stage, which can then guide the decoding process and help reduce computational costs. In contrast, other attention heads exhibit attention distributions that vary significantly across decoding steps, maintaining a broader attention scope without focusing on specific tokens at each step. For these heads, it is crucial to retain most of the KV pairs to ensure accurate predictions. For clarity, we provide the further claims for these two types of attention heads:

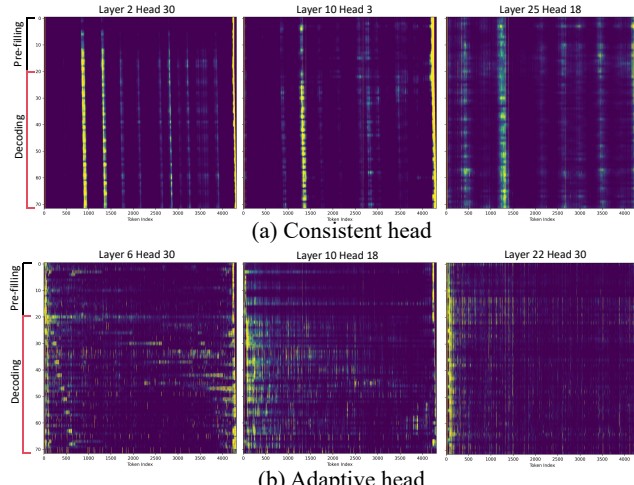

(a) Consistent head

(b) Adaptive head

Figure 2: Visualization of attention weight heatmaps for consistent heads and adaptive heads in the LongChat-7B model using a randomly selected text segment. In these figures, the vertical axis represents the attention of the head across different queries, with the first 20 rows corresponding to the last 20 queries during the pre-filling stage, while the subsequent rows depict the attention weights across consecutive decoding steps. These two types of heads exist in various layers and other popular models (Appendix A).

- **Consistent heads** are attention heads that exhibit stable attention patterns of sequential consistency, focusing on a limited set of tokens across all decoding steps.

- **Adaptive heads** are attention heads characterized by variable attention distributions across decoding steps, which do not exhibit stable patterns and require a larger attention space for flexible token interactions.

### 3.2. How to Identify Consistent and Adaptive Heads?

**Observation matrix** As can be seen in Figure 2, the attention weights of the last query tokens at the pre-filling stage effectively reflect the attention patterns. To save the computational cost, we set an observation window that contains the last query tokens of the input to identify head types and critical tokens. Additionally, since initial and recent tokens are typically important but do not influence the classification process, we temporarily exclude them during the identification phase. Formally, let $L_{\text{init}}$, $L_{\text{rec}}$ and $L_{\text{obs}}$ respectively denote the lengths of the initial tokens, recent tokens, and the observation window. Then, consider a head within a sample at the pre-filling stage, where the input includes the query $Q$, key $K$, and value $V \in \mathbb{R}^{n \times d}$, with $n$ representing the input length and $d$ representing the head dimension in the attention mechanism. We define the observation matrix

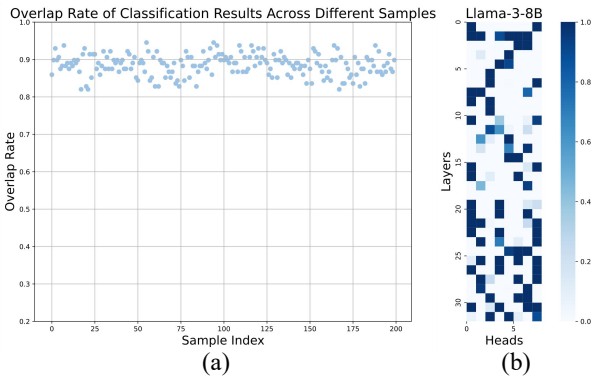

(a)           (b)

Figure 3: (a) High similarity of classification results across samples. We calculate the overlap rate between the classification results of 200 randomly selected WikiText samples and the overall classification result with $r = 0.3$. All samples exhibit an overlap rate exceeding 80% with the overall result. (b) Frequency of adaptive head identification for Llama-3B model in a reference dataset with $r = 0.3$. The frequency distribution is highly concentrated, enabling the determination of a fixed head type based on these results.

$A$ within the observation window, as follows:

$$A = \text{softmax}\left(\frac{Q_{[-L_{\text{obs}}:,:]} K_{[L_{\text{init}}:-L_{\text{rec}},:]}^T}{\sqrt{d}}\right), \quad (1)$$

where $A \in \mathbb{R}^{L_{\text{obs}} \times (n - L_{\text{rec}} - L_{\text{init}})}$. In the following, we will identify consistent heads based on the characteristics of $A$.

**Coefficient of Variation (CV) Score** Empirically, consistent heads exhibit two primary features in their observation matrices: a high degree of similarity among different rows and a small subset of columns that is sufficient to recall most of the attention weights. Inspired by this aspect, we propose a coefficient-of-variation-based score to measure the concentration and similarity of attention within observation matrices $A$. Since the coefficient of variation (Abdi, 2010) is highly sensitive to the magnitude of values, we first binarize the observation matrix with a percentile-based threshold $k$ and a scaling factor $\alpha$.

$$B = \mathbb{I}(A \geq \Phi(k, \alpha)) \in \mathbb{R}^{L_{\text{obs}} \times (n - L_{\text{rec}} - L_{\text{init}})} \quad (2)$$

where $\mathbb{I}$ is the indicator function, and $\Phi(k, \alpha) = \text{Quantile}_k(A) \times \alpha$ represents the $k$-th percentile element in the matrix $A$, scaled by a factor of $\alpha$. Then after binarization, we derive a frequency vector $C$ to quantify the number of times each token that is identified as critical, reflecting the similarity and concentration of attention weights:

$$C = \sum_{i=0}^{L_{\text{obs}}} B_{i,:} \in \mathbb{R}^{(n - L_{\text{rec}} - L_{\text{init}})} \quad (3)$$

---

**Algorithm 1** CateKV in an individual Head

**Input:** $Q, K, V \in \mathbb{R}^{n \times d}$, $q \in \mathbb{R}^{1 \times d}$, head type $H$, observation window size $L_{\text{obs}}$, selected chunk budget $b$, chunk size $cs$, retention ratio $\eta$
**Pre-filling Stage:**

*# Calculating token criticality*
$A = \text{softmax}(Q_{[-L_{\text{obs}}:,:]} K_{[:-L_{\text{obs}},:]}^T / \sqrt{d})$
$C = \sum_{i=0}^{L_{\text{obs}}} A_{i,:}$
*# Divide C into chunks and take maximum*
$C_{\text{chunk}} = \text{BlockMax}(C, cs)$
*# Cache keys and values based on the indices of top-k elements*
**if** $H = $ consistent head **then**
    $i_k = \text{argtopk}(C_{\text{chunk}}, b)$
    $cache_k, cache_v = \text{Cache}(K, V, i_k, L_{\text{obs}})$
**else**
    $i_k = \text{argtopk}(C_{\text{chunk}}, n\eta)$
    $cache_k, cache_v = \text{Cache}(K, V, i_k, L_{\text{obs}})$
**end if**
**Decoding Stage:**

*# Retrieval keys and values from cache*
**if** $H = $ consistent head **then**
    $k, v = cache_k, cache_v$
**else**
    $k, v = \text{Retrival}(q, cache_k, cache_v)$ *# all or query-aware*
**end if**
$output = \text{Attention}(q, k, v)$

---

Now, we can obtain the score based on the coefficient of variation for an attention head of a sample as follows:

$$\text{score} = \frac{\sqrt{\frac{1}{(n - L_{\text{rec}} - L_{\text{init}})} \sum_i (C_i - \mu(C))^2}}{\mu(C)} \quad (4)$$

where $\mu(C) = \frac{1}{(n - L_{\text{rec}} - L_{\text{init}})} \sum_i C_i$ is the mean. With the above equations, we can compute a score for each head under a sample. Then, for a specific LLM, we can obtain a score matrix $S \in \mathbb{R}^{l \times h}$ for all heads, where $l$ and $h$ represent the number of layers in the model and the number of heads in a layer, respectively. For a statistic head identification rule, let $r$ denote the proportion of adaptive heads, and $\Gamma(r)$ represents the percentile threshold based on $r$. We can use the threshold to control the token eviction ratio. Then, we distinguish the head types of a specific sample as follows

$$\text{Head}_{i,j} = \begin{cases} \text{consistent head,} & \text{if } S_{i,j} > \Gamma(r) \\ \text{adaptive head,} & \text{if } S_{i,j} \leq \Gamma(r) \end{cases} \quad (5)$$

**Reference-Based Static Identification** Although head identification can be performed dynamically for each sample, it is actually expensive for memory management along with the change of samples at the pre-filling stage. Therefore, it is better to determine a fixed head type for the model, which can be directly used for inference. Surprisingly, we

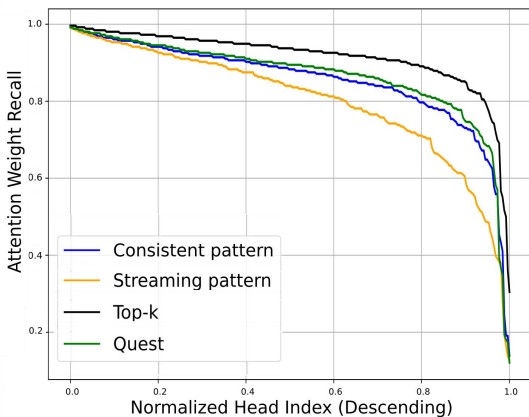

Figure 4: Attention weight recall curves for four methods. The curves show the attention recall for each attention head, calculated on four patterns: consistent, streaming, quest, and top-k, on a 128k-length example. The attention recall values are obtained by the mean of all decoding steps and sorted in descending order for all heads. The sparse budget is set to 2048, and the chunk size is 8. The results indicate that the consistent pattern outperforms the streaming pattern in terms of overall attention recall, and approaches the performance of the quest pattern that requires additional computation.

observe that the same head does exhibit similar attention patterns across different samples, and the identification results based on CV scores are highly consistent. This inspires us to use a reference dataset to calculate the frequency of each head being identified as the consistent head, and then derive the final model-wise identification result based on the adaptive head ratio $r$, as shown in Figure 3, which comes to the final form of our method (termed as *CateKV*). After determining the type of each head, we retain only the most important KV pairs for consistent heads, while for adaptive heads, we preserve a majority based on the predefined retention ratio $\eta$. Specifically, we select the top-$k$ KV pairs in chunks, enabling seamless integration with retrieval-based methods. For the GQA model, the observation matrix $A$ is computed as the mean of $A$ of the heads in a group. The CateKV acceleration for LLMs is shown in Algorithm 1.

### 3.3. Theory Analysis

In this part, we present an analysis of the theoretical bound on the eviction performance of CateKV.

**Lemma 1:** Let $G$ represent the hypothesis class derived from the CV-based function, $F$ denote the real-valued function class induced by the binary cross-entropy loss applied to $G$, and let $N$ denote the sample size of the reference-based dataset. Then, with probability at least $1 - \delta$, the following Rademacher complexity bound holds:

$$\forall f \in F, P_{\text{head}} \left( \mathbb{E}[f] - \frac{1}{N} \sum_{n=1}^{N} f_n \le 2\mathcal{R}_N(F) \right)$$
$$+ \sqrt{\frac{2 \log \frac{2}{\delta}}{2N}} \ge 1 - \delta. \tag{6}$$

where $\mathcal{R}_N(F)$ is the conditional Rademacher average.

Let $P_1$ and $P_2$ denote the probabilities of correctly classified consistent heads and adaptive heads, respectively, while $\bar{P}_1$ and $\bar{P}_2$ represent the probabilities of misclassified consistent heads and adaptive heads, respectively. It is assumed that $P_1 + P_2 = P_{\text{head}}$ and $\bar{P}_1 + \bar{P}_2 = 1 - P_{\text{head}}$. The probability in the above lemma can then be decomposed through a fine-grained analysis as follows.

**Theorem 1:** Let $\eta_1$ and $\eta_2$ denote the retention ratios for consistent heads and adaptive heads, respectively, while $\eta_1^*$ and $\eta_2^*$ represent their corresponding optimal retention ratios. Define the retention accuracy for different cases $r_{i,j} = \eta_i^* \mathbb{1}[\eta_j > \eta_i^*] + \eta_j(1 - \mathbb{1}[\eta_j > \eta_i^*])$ by comparing the retention budgets with the optimal budgets. Additionally, assume that the hypothesis asserting the query attention score provides the best description in order with probability $\lambda$. Then, the token retention accuracy satisfies:

$$P_{\text{token}} = \lambda \left( r_{1,1}P_1 + r_{2,2}P_2 + r_{2,1}\bar{P}_1 + r_{1,2}\bar{P}_2 \right)$$
$$\ge \lambda \left( \min(r_{2,1}, r_{1,2}) + [\min(r_{1,1}, r_{2,2}) \right.$$
$$\left. - \min(r_{2,1}, r_{1,2})] P_{\text{head}} \right). \tag{7}$$

**Remark 1:** From the above theorem, three critical factors influence the worst-case token retention accuracy (i.e., the lower bound):

- $\lambda$: The effectiveness of identifying an efficient measure to characterize token correlation by the score order, with as high a probability of correctness as possible.

- Budget control: The ability to appropriately set the retention budget in order to maximize gains by reducing the majority of tokens when heads are correctly classified, while simultaneously mitigating the negative impact when heads are misclassified.

- $P_{\text{head}}$: The accuracy of the CV-based method in classifying the head type during token reduction, which directly impacts the overall performance.

### 3.4. Further Discussion of CateKV

To show the distinction and effectiveness of attention patterns discovered from the sequential consistency, Figure 4

Table 1: Performance (%) of different models and various methods on RULER evaluated at length of 128K. The 'Cache' in the table refers to the retained KV cache size. CateKV outperforms other methods and comparable with full attention.

| Methods | Cache | N-S1 | N-S2 | N-S3 | N-MK1 | N-MK2 | N-MK3 | FWE | N-MQ | N-MV | QA-1 | QA-2 | VT | Avg. |
|---|---|---|---|---|---|---|---|---|---|---|---|---|---|---|
| *Llama-3-8B-1M* | 100% | 100.00 | 100.00 | 100.00 | 98.96 | 98.96 | 41.67 | 71.88 | 98.69 | 96.35 | 73.96 | 50.00 | 78.75 | 84.10 |
| StreamingLLM | 41% | 51.04 | 51.04 | 51.04 | 40.63 | 35.42 | 22.92 | 75.69 | 45.31 | 39.58 | 78.13 | 45.83 | 31.46 | 47.34 |
| SnapKV | 41% | 100.00 | 100.00 | 100.00 | 98.96 | 98.96 | 30.21 | 71.53 | 98.44 | 97.13 | 73.95 | 51.04 | 79.17 | 83.28 |
| PyramidKV | 41% | 100.00 | 100.00 | 100.00 | 98.96 | 98.96 | 37.50 | 71.53 | 98.44 | 96.61 | 71.87 | 50.00 | 79.38 | 83.60 |
| Duoattention | 41% | 100.00 | 100.00 | 100.00 | 98.96 | 97.92 | 39.58 | 76.74 | 94.27 | 90.36 | 69.79 | 51.04 | 86.46 | 83.76 |
| CateKV | 41% | 100.00 | 100.00 | 100.00 | 98.96 | 97.92 | 41.67 | 71.88 | 98.44 | 96.88 | 73.96 | 50.00 | 85.63 | 84.61 |
| *Phi-3-Mini-128K* | 100% | 96.88 | 90.63 | 95.83 | 83.33 | 65.63 | 37.50 | 87.15 | 72.14 | 66.67 | 63.54 | 39.58 | 65.83 | 72.06 |
| StreamingLLM | 41% | 47.91 | 45.83 | 44.79 | 38.54 | 30.21 | 25.00 | 84.38 | 36.49 | 34.90 | 64.58 | 38.54 | 4.38 | 41.29 |
| SnapKV | 41% | 96.88 | 90.63 | 80.21 | 82.29 | 56.25 | 11.46 | 82.99 | 61.72 | 53.91 | 62.50 | 38.54 | 63.54 | 65.08 |
| PyramidKV | 41% | 96.88 | 90.63 | 84.38 | 83.33 | 57.29 | 13.54 | 78.47 | 66.15 | 59.64 | 62.50 | 39.58 | 62.29 | 66.22 |
| CateKV | 41% | 96.88 | 90.63 | 95.83 | 83.33 | 65.63 | 38.54 | 80.21 | 70.31 | 65.63 | 63.54 | 39.58 | 70.21 | 71.69 |
| *Llama-3.1-8B* | 100% | 100.00 | 100.00 | 98.96 | 98.96 | 90.63 | 63.54 | 71.53 | 98.96 | 95.31 | 81.25 | 46.88 | 68.54 | 84.55 |
| StreamingLLM | 41% | 51.04 | 51.04 | 51.04 | 39.58 | 34.38 | 40.63 | 71.18 | 44.27 | 39.84 | 85.42 | 40.63 | 28.33 | 48.11 |
| SnapKV | 41% | 100.00 | 100.00 | 98.96 | 98.96 | 89.58 | 46.88 | 69.10 | 98.96 | 94.01 | 81.25 | 46.88 | 68.96 | 82.80 |
| PyramidKV | 41% | 100.00 | 100.00 | 98.96 | 98.96 | 90.63 | 56.25 | 65.28 | 98.96 | 95.31 | 80.21 | 46.88 | 65.42 | 83.07 |
| Duoattention | 41% | 100.00 | 100.00 | 98.96 | 97.92 | 88.54 | 59.38 | 74.31 | 97.92 | 91.41 | 81.25 | 46.88 | 78.54 | 84.59 |
| CateKV | 41% | 100.00 | 100.00 | 98.96 | 98.96 | 88.54 | 61.46 | 71.88 | 98.96 | 94.27 | 81.25 | 46.88 | 74.79 | 84.66 |
| *Yi-9B-200K* | 100% | 100.00 | 100.00 | 98.96 | 85.42 | 63.54 | 18.75 | 89.24 | 66.41 | 32.55 | 45.83 | 38.54 | 35.00 | 64.52 |
| StreamingLLM | 41% | 47.92 | 52.08 | 50.00 | 39.58 | 37.50 | 7.29 | 90.28 | 33.33 | 14.84 | 44.79 | 36.46 | 18.13 | 39.35 |
| SnapKV | 41% | 100.00 | 100.00 | 95.83 | 85.41 | 43.75 | 3.13 | 83.33 | 66.41 | 33.33 | 46.88 | 40.63 | 38.96 | 61.47 |
| PyramidKV | 41% | 100.00 | 100.00 | 91.67 | 86.46 | 50.00 | 2.08 | 73.61 | 68.75 | 34.64 | 43.75 | 37.50 | 41.46 | 60.83 |
| CateKV | 41% | 100.00 | 100.00 | 100.00 | 84.38 | 70.83 | 18.75 | 92.01 | 62.24 | 34.64 | 43.75 | 37.50 | 45.00 | 65.76 |

compares attention weight recall for our consistent pattern, streaming pattern (Xiao et al., 2023), Quest pattern (Tang et al., 2024b), and the upper-bound top-k pattern under the same sparse budget. As can be seen, all methods exhibit a gap compared to the upper bound, indicating some information loss with current sparse attention approaches. Therefore, retaining most KV pairs for certain heads is important for maintaining accuracy. On the other hand, the attention recall of the consistent pattern closely approximates that of the Quest pattern, applying the consistent pattern to heads with sequential consistency can help KV retrieval methods reduce memory usage and the cost of selecting critical tokens. Related works (Tang et al., 2024a; Xiao et al., 2024b) classify attention heads into Retrieval Heads and Streaming Heads, which is similar to our method. However, from an attention recall perspective, the streaming pattern is only effective for a small fraction of heads, otherwise deviating significantly from full attention. This highlights that the consistent pattern achieves a higher compression rate than the streaming pattern. We will discuss the comparison and integration with these methods in the experimental section.

## 4. Experiments

### 4.1. Setup

**LLM and Benchmark** We employed five widely used LLMs in long-context scenarios: LLaMA-3-8B-Instruct-1048K (Gradient, 2024a), Phi-3-Mini-128K (Abdin et al., 2024), Llama-3.1-8B (Meta AI, 2024), Yi-9B-200K (AI

et al., 2024) and Qwen2.5-7B (Bai et al., 2023a). The performance of CateKV was assessed on three challenging benchmarks: RULER (Hsieh et al., 2024), LongBench (Bai et al., 2023b), and Needle in a Haystack (NIAH) (Kamradt, 2024). We built a reference set for head identification of CateKV by emulating Variable Tracking task from RULER, which is very distinct from the test set. The experiments were carried out on a single NVIDIA A100-80G GPU.

**Baselines** We compare CateKV with eviction-based algorithms StreamingLLM (Xiao et al., 2023), SnapKV (Li et al., 2024), PyramidKV (Cai et al., 2024), retrieval-based algorithms Quest (Tang et al., 2024b) and ShadowKV (Sun et al., 2024), and the head-wise classification algorithm Duoattention (Xiao et al., 2024b). In our experiments, all approaches maintained an exact pre-filling stage and utilized sparse attention during the decoding stage. We also do not perform memory optimization like ShadowKV in the accuracy comparison. For fairness, when comparing with the KV eviction methods and head classification methods, we maintain the same KV cache size, while comparing with KV retrieval methods, we integrate them under adaptive heads to ensure an equivalent computational budget. Given that Duoattention only provides attention patterns for Llama3 and Llama3.1 in the models used in our study, the comparison is restricted to these two models. When baselines require a chunk size, we all set it to 8 to maintain consistency. Further experimental details are in the Appendix B.1.

Table 2: Performance (%) of different models and various methods on LongBench. We present the average score of all 21 tasks. The "Budget" refers to the computational budget. Please refer to the Appendix B.3 for detailed data.

| Model | | | LLama-3 | Phi-3 | Llama-3.1 | Yi | Qwen2.5 |
|---|---|---|---|---|---|---|---|
| Methods | Cache | Budget | | | Average Performance | | |
| Full | 100% | 100% | 31.27 | 34.00 | 33.68 | 33.02 | 30.03 |
| +CateKV | 42% | 42% | 31.48 | 33.73 | 33.70 | 32.83 | 29.92 |
| Quest | 100% | 3% | 30.90 | 33.11 | 33.20 | 32.82 | 29.30 |
| +CateKV | 42% | 3% | 31.03 | 33.29 | 33.38 | 32.79 | 29.33 |
| ShadowKV | 100% | 3% | 30.77 | 32.53 | 33.03 | 32.41 | 28.57 |
| +CateKV | 42% | 3% | 30.94 | 32.45 | 32.96 | 32.21 | 28.51 |

## 4.2. Effectiveness Evaluation

### 4.2.1. RESULTS ON RULER

In this experiment, we test 12 synthetic tasks under the context of 128K, with each task including 96 samples. To ensure a fair comparison with other baseline methods, we focus here on the task-aware setting. The results are shown in Table 1. Specifically, in CateKV , we set the adaptive head ratio $r$ to 0.4, the retention ratio $\eta$ to 1.0 and allocate a sparse budget for consistent heads at 2048 (1.56%), retaining approximately 41% of the KV cache size. Experimental results demonstrate that our method outperforms baselines and is comparable to full attention, despite evicting over half of the KV pairs. Besides, CateKV exhibits outstanding performance in complex tasks such as multi-document QA and variable tracking while maintaining high accuracy in other tasks. Due to the space limitation, we place results of more context lengths and combination with other retrieval-based methods in Appendix B.2.

### 4.2.2. RESULTS ON LONGBENCH

LongBench (Bai et al., 2023b) is a comprehensive long-context benchmark including 6 main categories and 21 diverse tasks. Following ShadowKV (Sun et al., 2024), we test samples with lengths exceeding 4096 tokens. In CateKV, we set the adaptive ratio $r$ and retention ratio $\eta$ to 0.4 and 1.0 respectively. The budget for consistent heads is set to 512. As shown in Table 2, CateKV enables a reduction in KV cache size to 42% across five LLMs while maintaining an accuracy decline of no more than 0.3% on average. Furthermore, when integrated with KV retrieval methods like Quest (Tang et al., 2024b) and ShadowKV (Sun et al., 2024), CateKV facilitates a reduction in memory usage with only a minimal impact on accuracy, not exceeding a 0.2% decrease, under the same computational budget.

### 4.2.3. RESULTS ON NEEDLE IN A HAYSTACK

As shown in Figure 5, on the Needle In A Haystack dataset, CateKV demonstrates its robust ability to accurately identify and extract relevant information from long contexts, ranging from 20K to 1M tokens, while reducing memory and

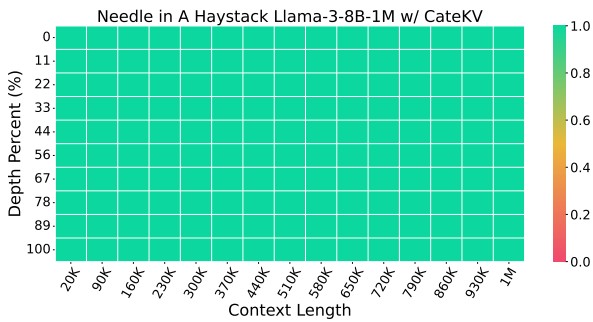

Figure 5: Needle In A Haystack from 20K to 1M in Llama3

Table 3: Performance of CateKV combined with Duoattention and MInference on the RULER benchmark. The results are tested on the Llama-3-8B-1M model.

| Methods | 8K | 16K | 32K | 64K | 128K | 256K | Avg. |
|---|---|---|---|---|---|---|---|
| Duoattention | 78.32 | 77.93 | 69.77 | 66.27 | 60.37 | 58.07 | 68.46 |
| Duoattention w/ CateKV | 89.93 | 91.08 | 89.96 | 86.44 | 84.57 | 81.59 | 87.26 |
| MInference | 91.33 | 92.28 | 89.66 | 84.97 | 84.57 | 81.10 | 87.32 |
| MInference w/ CateKV | 91.32 | 92.13 | 89.71 | 85.90 | 85.14 | 82.27 | 87.74 |

computational cost by more than half. Additional NIAH test results on other models are available in the Appendix B.4.

### 4.2.4. CATEKV VS. DUOATTENTION

Duoattention (Xiao et al., 2024b) classifies attention heads into retrieval heads and streaming heads, similar to our approach. For a comprehensive comparison, we evaluate CateKV against DuoAttention across varying ratios of adaptive/retrieval heads, ranging from 0.1 to 1.0. Figure 7 shows that Duoattention's accuracy notably declines when the proportion of retrieval heads falls below 0.3. In contrast, CateKV sustains performance close to full attention even when the ratio of adaptive heads decreases to 0.1. This discrepancy arises from the fundamental difference in attention patterns between our consistent head and their streaming head. The consistent head in CateKV captures similarity across different queries, whereas the streaming head primarily focuses on the coverage of initial and recent tokens. This essential difference enables our method to complement DuoAttention, mitigating its performance degradation at low full attention head ratios. To validate this, we conducted experiments presented in Table 3. Specifically, we set the retrieval head ratio in DuoAttention to 0.3, with the remaining 0.7 as streaming heads. In contrast, "Duoattention w/ CateKV" replaces part of the streaming heads with consistent heads, resulting in a 0.2 consistent head ratio and a 0.5 streaming head ratio. The results demonstrate that incorporating CateKV significantly improves DuoAttention's performance at low full attention head ratios, further validating the effectiveness of our consistency pattern.

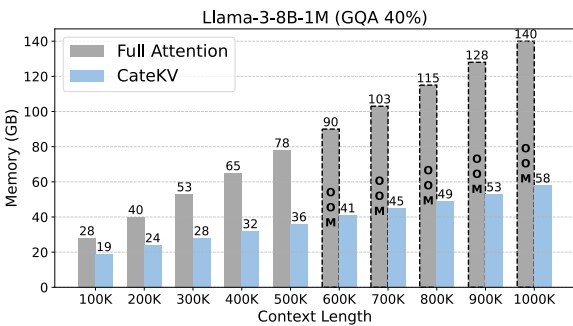 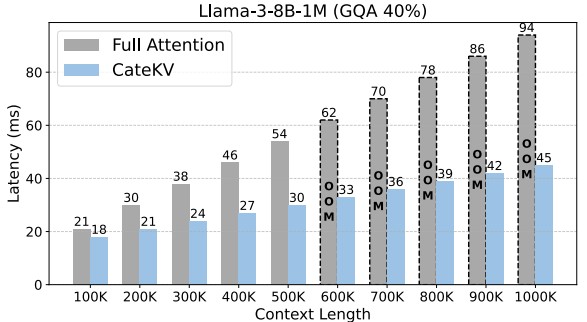

Figure 6: Comparison of decoding efficiency between CateKV and full attention. As the context length increases, both the memory usage (a) and decoding latency (b) increase linearly, but CateKV exhibits a smaller slope compared to full attention. OOM indicates that the GPU memory limit (80G) is exceeded; corresponding data is obtained by extrapolation.

### 4.2.5. EFFICIENT PRE-FILLING METHODS INTEGRATION

CateKV is designed to accelerate the decoding stage and can naturally integrate with prefilling acceleration methods. We also combined CateKV with the efficient prefilling acceleration method, MInference (Jiang et al., 2024), and tested it on RULER with contexts ranging from 8K to 256K. As shown in Table 3, combining CateKV with MInference does not lead to performance degradation and even a slight performance improvement has been observed. This shows that sequential consistency within the heads is unaffected by the pre-filling inference patterns, highlighting the robustness of our method. Moreover, the additional acceleration of the prefilling stage further improves the overall inference speed.

### 4.3. Efficiency Evaluation

To validate the efficiency of CateKV, we tested it under both single-sample and batch-sample inference scenarios.

### 4.3.1. SINGLE-SAMPLE INFERENCE

To evaluate the efficiency of CateKV during single-sample inference, we selected three models with different numbers of KV heads: Llama-3, Phi-3, and Yi. We compared CateKV to full attention on these models by measuring memory usage and decoding latency at the same input length. We observed that memory and latency reductions increase as the context length grows (detailed in the Figure 6). To demonstrate optimal performance, we selected the maximum input length that can be handled by full attention on a single A100 GPU. As shown in Table 4, under the generic settings of $r = 0.4$ and $\eta = 1.0$, the Phi-3 model achieved reductions of $2.11\times$ in memory and $1.79\times$ in latency by using CateKV. By balancing efficiency and accuracy, CateKV further reduced memory usage by $2.72\times$ and decoding latency by $2.18\times$ on Llama-3, with accuracy decline on RULER-128K and Longbench tasks under $0.25\%$.

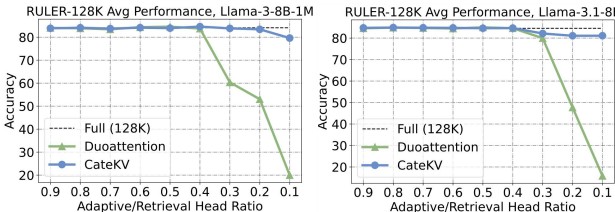

Figure 7: Comparison of accuracy between our method and DuoAttention across different full attention head ratios.

### 4.3.2. BATCH-SAMPLE INFERENCE

Batch-sample inference is a more common scenario in real-world applications. By evicting a portion of the KV pairs, CateKV supports larger batch sizes, thereby increasing throughput. To evaluate CateKV's efficiency in batch-sample scenarios, we set each sample length to 40K and input them in batches, comparing throughput at the maximum batch size for both full attention and CateKV. The comparative results across three models are presented in Table 4. With the generic settings, gains of up to $2.38\times$ in batch size and $2.25\times$ in throughput were achieved. Additionally, through further KV cache compression while maintaining accuracy, both batch size and throughput in the Llama-3 model were boosted to $4.33\times$ and $3.96\times$, respectively.

### 4.4. Ablation Study

The ablation studies focus on three key aspects of CateKV : (1) ratio of adaptive heads, (2) retention ratio in the adaptive head, and (3) sparse KV cache budget in the consistent head. All experiments are conducted on the RULER-128K datasets using the Llama-3-8B-1M model.

### 4.4.1. RATIO OF ADAPTIVE HEADS

The ratio of adaptive heads $r$ is a crucial hyperparameter balancing accuracy and inference speed. As shown in Figure 7, the relationship between $r$ and model performance does

Table 4: Efficiency comparison between CateKV and Full Attention on a single A100 GPU. In the single-sample inference task, we utilized texts of lengths 500K, 180K, and 650K as inputs for the following three models. For the batch-sample inference task, the sample length was set at 40K, and the maximum feasible batch size was used for each method evaluated.

| | Accuracy (Avg) | | Single-sample inference | | Batch-sample inference | |
|---|---|---|---|---|---|---|
| Model | RULER-128K | Longbench | Memory | Latency | Batchsize | Throughput |
| *Llama-3-8B-1M* (8 KV heads) | 84.10 (0.00) | 31.27 (0.00) | 77.72 (1.00×) | 54.28 (1.00×) | 12 (1.00×) | 229.37 (1.00×) |
| CateKV ($r = 0.4, \eta = 1.0$) | 84.61 (+0.51) | 31.48 (+0.21) | 41.02 (1.89×) | 32.75 (1.66×) | 28 (2.33×) | 511.46 (2.23×) |
| CateKV ($r = 0.3, \eta = 0.7$) | 83.85 (-0.25) | 31.21 (-0.06) | 28.59 (2.72×) | 24.86 (2.18×) | 52 (4.33×) | 909.69 (3.96×) |
| *Phi-3-Mini-128K* (32 KV heads) | 72.06 (0.00) | 34.00 (0.00) | 75.11 (1.00×) | 55.51 (1.00×) | 4 (1.00×) | 78.53 (1.00×) |
| CateKV ($r = 0.4, \eta = 1.0$) | 71.69 (-0.37) | 33.73 (-0.27) | 35.58 (2.11×) | 31.06 (1.79×) | 10 (2.50×) | 167.11 (2.13×) |
| CateKV ($r = 0.3, \eta = 1.0$) | 71.48 (-0.58) | 33.66 (-0.33) | 29.02 (2.59×) | 27.81 (2.00×) | 14 (3.50×) | 221.29 (2.82×) |
| *Yi-9B-200K* (4 KV heads) | 64.52 (0.00) | 33.02 (0.00) | 77.65 (1.00×) | 56.30 (1.00×) | 16 (1.00×) | 292.55 (1.00×) |
| CateKV ($r = 0.4, \eta = 1.0$) | 65.76 (+1.24) | 32.83 (-0.19) | 41.62 (1.87×) | 36.05 (1.56×) | 38 (2.38×) | 659.02 (2.25×) |
| CateKV ($r = 0.3, \eta = 0.8$) | 65.75 (+1.23) | 32.78 (-0.24) | 31.58 (2.46×) | 29.96 (1.88×) | 60 (3.75×) | 980.86 (3.35×) |

Figure 8: (a) The effect of adaptive head ratio on memory and decoding latency is approximately linear. (b) Impact of retention ratio on accuracy in RULER-128K and decoding latency for 500K length input. (c) Minimal effect of sparse budget in consistent heads on accuracy.

not exhibit a perfect inverse correlation. Rather, the performance remains relatively stable with decreasing $r$ until reaching a critical threshold (≈0.2), beyond which significant degradation occurs due to excessive reliance on critical tokens during the prefilling stage. In terms of efficiency, as shown in Figure 8(a), both memory usage and decoding latency decrease almost linearly with decreasing $r$.

### 4.4.2. RETENTION RATIO IN ADAPTIVE HEAD

In adaptive heads, there is still a subset of tokens that are always not important. This enables a reduction in the proportion of the KV cache retained within adaptive heads. Figure 8(b) illustrates the impact of changes in the retention ratio $\eta$ on accuracy and latency. When $\eta$ exceeds a certain threshold (≈0.6), the model maintains or even surpasses the performance achieved with the full KV cache. However, dropping below this threshold results in a significant performance decline, indicating that adaptive heads heavily depend on the majority of the KV cache. Additionally, a decrease in $\eta$ leads to a linear reduction in latency. In practical applications, it is essential to adjust both $r$ and $\eta$ to balance accuracy and memory consumption.

### 4.4.3. SPARSE BUDGET IN CONSISTENT HEAD

As illustrated in Figure 8(c), CateKV demonstrates strong robustness across different sparse budgets. Under the set-

tings of $r = 0.4$ and $\eta = 1.0$, even when the sparse budget in the consistent head is reduced to approximately 0.78% (1024), CateKV still maintains comparable performance to full attention in terms of average accuracy. This indicates that, during inference, the consistent head only requires a minimal portion of the cache to perform its function, while the acquisition of global information relies primarily on adaptive heads, which retain the majority of KV pairs during the decoding stage. Since the sparse budget for the consistent head is a small fraction of the total cache, its impact on memory usage and inference latency is negligible.

## 5. Conclusion

We propose CateKV, a novel hybrid KV cache method that leverages sequential consistency to improve LLM inference efficiency in long-context tasks. By using a coefficient-of-variation-based algorithm, CateKV classifies attention heads into consistent and adaptive types. It selectively retains critical KV pairs in consistent heads and most pairs in adaptive heads, reducing memory usage and decoding latency while maintaining performance. Additionally, it can be easily integrated with other acceleration methods for further enhancement. Extensive evaluations demonstrate that CateKV achieves significant efficiency gains, including up to 2.72× reduction in memory usage, 2.18× acceleration in decoding, and a 3.96× throughput increase in batch scenarios.

## Acknowledgments

This work is supported by National Natural Science Foundation of China (No.62306178), STCSM (No.22DZ2229005), 111 plan (No. BP0719010).

## Impact Statement

The primary objective of our research is to develop inference acceleration methods for large language models, aiming to advance their efficiency and scalability. We explicitly state that any depictions of violence in the datasets are wholly fictional and are utilized exclusively for academic research purposes. This work does not reflect, support, or justify any real-world violent actions. Additionally, this research was carried out independently, with no external funding or conflicts of interest influencing its outcomes. The study strictly follows ethical guidelines, taking into account essential factors such as discrimination, bias, fairness, privacy, security, and legal adherence, while maintaining the utmost integrity in the research process.

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

# A. Further Observations

We expanded our exploration of sequential consistency to a wider selection of models, primarily focusing on the popular models mentioned in the main text: Llama-3-8B-1M (Gradient, 2024a), Llama-3.1-8B (Meta AI, 2024), Phi-3-128K (Abdin et al., 2024), and Yi-9B-200K (AI et al., 2024). In Figure 9, we visualized the attention weights heatmap for these four models, illustrating the presence of both consistent and adaptive heads across various layers. This visualization supports the generality of our observations. Consistent with the setup described in the main text, the heatmap in the figure comprises attention weights associated with the last 20 query tokens during the pre-filling stage and all query tokens during the decoding stage, employing a causal mask. And the samples are randomly excerpted from the WikiText-2 (Merity et al., 2016) dataset.

# B. Experiment Details

## B.1. Implementation Details of Experiments

During the identification stage of CateKV, we employed an observation window and temporarily excluded initial tokens and recent tokens from the context window. We set $L_{obs}$ to 64, while $L_{init}$ and $L_{rec}$ were defined as 1/32 and 1/128 of the sparse budget, respectively. We constructed the reference dataset based on the Variable Tracking task from the RULER Benchmark, which comprises 100 samples, each with a length of 128K, distinct from the test set. The sparse budget was set at 2048. According to the performance on the reference dataset, we selected the most appropriate percentile threshold $k$ and scaling factor $\alpha$ for each model. For the percentile threshold $k$, Llama3 and Llama3.1 were set at 0.996 and 0.984, respectively, while other models were set at 0.99. For the scaling factor $\alpha$, Llama3.1, and Yi were set at 0.8, while other models were assigned a value of 1.0.

For the baseline methods, we configured the observation windows of SnapKV (Li et al., 2024) and PyramidKV (Cai et al., 2024) to 32 and set the $\beta$ in PyramidKV to 20. For StreamingLLM (Xiao et al., 2023), the initial tokens were set to 128. Regarding Duoattention(Xiao et al., 2024b), we conducted experiments using the attention patterns provided by their code available on GitHub.

## B.2. Additional Results on RULER

### B.2.1. PERFORMANCE OF DIFFERENT CONTEXT LENGTHS ON RULER

We also conducted evaluations on various context lengths within the RULER benchmark. The Table 5 presents the performance of CateKV on tasks with context lengths from 8K to 256K. CateKV is comparable to full attention in terms of all lengths and average results and even shows slight improvements in performance at certain lengths when $r = 0.4$ and $\eta = 1.0$.

Table 5: Performance of different context lengths on RULER

| Methods | 8K | 16K | 32K | 64K | 128K | 256K | Avg. |
|---|---|---|---|---|---|---|---|
| *Llama-3-8B-1M* | 91.47 | 92.87 | 90.31 | 86.44 | 84.10 | 79.79 | 87.50 |
| CateKV | 91.28 | 92.11 | 90.37 | 86.86 | 84.61 | 81.53 | 87.79 |
| *Phi-3-Mini-128K* | 92.02 | 91.42 | 91.24 | 87.89 | 72.06 | - | 86.93 |
| CateKV | 92.68 | 92.54 | 92.04 | 88.78 | 71.69 | - | 87.55 |
| *Llama-3.1-8B* | 94.78 | 94.95 | 94.61 | 93.02 | 84.55 | - | 92.38 |
| CateKV | 94.66 | 94.68 | 94.58 | 93.03 | 84.66 | - | 92.32 |
| *Yi-9B-200K* | 87.54 | 82.33 | 72.06 | 69.19 | 64.52 | - | 75.13 |
| CateKV | 87.17 | 81.68 | 71.85 | 69.25 | 65.76 | - | 75.14 |

### B.2.2. COMBINE WITH KV RETRIEVAL METHODS

We combine CateKV with Quest (Tang et al., 2024b) and ShadowKV (Sun et al., 2024) and compare their performance with the baseline under the same computational budget at a length of 128K. The results are shown in Table 6. CateKV helps

Table 6: Performance (%) of CateKV combined with KV retrieval methods. 'Budget' refers to the computational budget for sparse attention. CateKV can reduce memory. CateKV can help KV retrieval methods reduce memory usage to 41% while maintaining accuracy.

| Methods | Budget | N-S1 | N-S2 | N-S3 | N-MK1 | N-MK2 | N-MK3 | FWE | N-MQ | N-MV | QA-1 | QA-2 | VT | Avg. |
|---|---|---|---|---|---|---|---|---|---|---|---|---|---|---|
| *Llama-3-8B-1M* | 100% | 100.00 | 100.00 | 100.00 | 98.96 | 98.96 | 41.67 | 71.88 | 98.69 | 96.35 | 73.96 | 50.00 | 78.75 | 84.10 |
| SnapKV | 1.6% | 100.00 | 100.00 | 14.58 | 98.96 | 96.88 | 0.00 | 61.11 | 98.44 | 96.88 | 68.75 | 48.96 | 79.38 | 72.00 |
| PyramidKV | 1.6% | 100.00 | 100.00 | 10.42 | 98.96 | 96.88 | 0.00 | 56.60 | 98.18 | 95.58 | 70.83 | 48.96 | 80.42 | 71.40 |
| Quest | 1.6% | 100.00 | 100.00 | 100.00 | 98.96 | 97.92 | 19.79 | 58.33 | 98.96 | 94.27 | 72.91 | 52.08 | 80.20 | 81.31 |
| CateKV+Quest | 1.6% | 100.00 | 100.00 | 100.00 | 97.92 | 97.92 | 19.79 | 58.33 | 98.96 | 94.27 | 73.96 | 51.04 | 82.29 | 81.21 |
| ShadowKV | 1.6% | 100.00 | 100.00 | 100.00 | 97.92 | 93.75 | 21.88 | 75.69 | 98.96 | 96.09 | 72.92 | 50.00 | 78.96 | 82.18 |
| CateKV+ShadowKV | 1.6% | 100.00 | 100.00 | 100.00 | 97.92 | 89.58 | 21.88 | 75.35 | 98.70 | 94.79 | 71.88 | 51.04 | 81.25 | 81.86 |
| *Phi-3-Mini-128K* | 100% | 96.88 | 90.63 | 95.83 | 83.33 | 65.63 | 37.50 | 87.15 | 72.14 | 66.67 | 63.54 | 39.58 | 65.83 | 72.06 |
| SnapKV | 1.6% | 98.21 | 38.54 | 1.04 | 42.71 | 11.46 | 0.00 | 60.76 | 8.59 | 2.60 | 62.50 | 38.54 | 60.63 | 35.47 |
| PyramidKV | 1.6% | 97.92 | 39.58 | 0.00 | 46.88 | 11.46 | 0.00 | 56.94 | 10.16 | 2.08 | 59.38 | 39.58 | 60.42 | 35.37 |
| Quest | 1.6% | 96.88 | 92.71 | 96.88 | 80.21 | 57.29 | 20.83 | 57.29 | 69.53 | 63.02 | 64.58 | 39.58 | 63.75 | 66.88 |
| CateKV+Quest | 1.6% | 96.88 | 92.71 | 96.88 | 80.21 | 57.29 | 18.75 | 55.56 | 68.75 | 64.32 | 64.58 | 39.58 | 62.50 | 66.50 |
| ShadowKV | 1.6% | 95.83 | 88.54 | 90.63 | 80.21 | 54.17 | 21.88 | 77.43 | 63.28 | 51.28 | 62.50 | 38.54 | 63.75 | 65.71 |
| CateKV+ShadowKV | 1.6% | 97.92 | 87.50 | 92.70 | 77.08 | 55.20 | 18.75 | 74.31 | 64.06 | 55.99 | 62.50 | 38.54 | 65.83 | 65.87 |
| *Llama-3.1-8B* | 100% | 100.00 | 100.00 | 98.96 | 98.96 | 90.63 | 63.54 | 71.53 | 98.96 | 95.31 | 81.25 | 46.88 | 68.54 | 84.55 |
| SnapKV | 1.6% | 100.00 | 100.00 | 41.67 | 98.96 | 79.17 | 0.00 | 59.72 | 97.14 | 91.67 | 81.25 | 44.79 | 62.92 | 71.44 |
| PyramidKV | 1.6% | 100.00 | 100.00 | 33.33 | 98.96 | 81.25 | 1.04 | 56.25 | 95.83 | 93.48 | 81.25 | 45.83 | 65.00 | 71.02 |
| Quest | 1.6% | 100.00 | 100.00 | 100.00 | 98.96 | 78.13 | 4.17 | 59.03 | 98.70 | 94.01 | 80.21 | 50.00 | 68.96 | 77.68 |
| CateKV+Quest | 1.6% | 100.00 | 100.00 | 100.00 | 98.96 | 78.13 | 3.13 | 62.15 | 97.92 | 90.63 | 82.29 | 47.92 | 66.88 | 77.33 |
| ShadowKV | 1.6% | 100.00 | 100.00 | 98.96 | 98.96 | 77.08 | 13.54 | 70.49 | 98.18 | 90.36 | 81.25 | 48.96 | 64.17 | 78.50 |
| CateKV+ShadowKV | 1.6% | 100.00 | 100.00 | 100.00 | 98.96 | 73.96 | 13.54 | 65.97 | 98.44 | 90.89 | 80.21 | 50.00 | 67.08 | 78.25 |
| *Yi-9B-200K* | 100% | 100.00 | 100.00 | 98.96 | 85.42 | 63.54 | 18.75 | 89.24 | 66.41 | 32.55 | 45.83 | 38.54 | 35.00 | 64.52 |
| SnapKV | 1.6% | 100.00 | 93.75 | 5.21 | 80.21 | 6.25 | 0.00 | 75.69 | 54.95 | 18.23 | 40.63 | 36.46 | 52.92 | 47.02 |
| PyramidKV | 1.6% | 100.00 | 94.79 | 4.17 | 79.17 | 5.21 | 0.00 | 85.76 | 55.73 | 15.63 | 41.67 | 33.33 | 51.45 | 47.24 |
| Quest | 1.6% | 100.00 | 97.92 | 98.96 | 85.42 | 64.58 | 4.17 | 67.01 | 66.14 | 38.39 | 42.71 | 36.46 | 48.33 | 62.51 |
| CateKV+Quest | 1.6% | 100.00 | 100.00 | 98.96 | 85.42 | 70.83 | 5.21 | 66.67 | 63.28 | 37.24 | 40.63 | 37.50 | 50.21 | 62.99 |
| ShadowKV | 1.6% | 100.00 | 100.00 | 97.92 | 87.50 | 60.42 | 2.08 | 75.35 | 59.64 | 34.11 | 43.75 | 37.50 | 54.79 | 62.76 |
| CateKV+ShadowKV | 1.6% | 100.00 | 100.00 | 97.92 | 84.38 | 58.33 | 2.08 | 82.29 | 59.38 | 33.85 | 43.75 | 36.46 | 47.29 | 62.14 |

reduce the memory usage of Quest and ShadowKV and significantly outperforms KV Eviction methods in terms of accuracy with the same computational load.

### B.3. Full Longbench Results

We present the complete experimental results for the Longbench in Table 7. We integrate CateKV with both the Full Attention and KV retrieval methods, Quest(Tang et al., 2024b) and ShadowKV(Sun et al., 2024), and evaluate its performance on all 21 tasks in Longbench. The results showed that this integration did not lead to any significant drop in per-task accuracy, and the average accuracy even outperformed the original methods, despite retaining only 42% of the KV cache size. For around half of the tasks, the combination of CateKV with the original methods leads to a slight improvement in performance.

### B.4. Additional Results in Needle In A Hystack

Figure B.4 displays the performance of the Llama3.1-8B, Phi-3-Mini-128K, and Yi-9B-200K models on the 'Needle In A Haystack' task. Compared to full attention, CateKV shows varied performance across different context windows and needle depths, but maintains overall comparable performance. This suggests that CateKV does not significantly affect the models' capacity to access and retrieve long-context semantic information.

### B.5. Additional Results on Larger Models

We evaluated the performance of CateKV on larger models, setting the context length according to the maximum supported by each model—128k for Qwen2.5-32B and Yi-34B-200K, and 16k for Phi-4-14B. As shown in Table 8, CateKV scales effectively, achieving near full-attention accuracy on the 30B and 14B models, outperforming baseline methods such as SnapKV and PyramidKV.

Table 7: Full LongBench results with Llama-3-8B-1M.

| Metrics | Full | CateKV | Quest | CateKV+Quest | ShadowKV | CateKV+ShadowKV |
|---|---|---|---|---|---|---|
| Average | 31.27 | **31.48** | 30.90 | **31.03** | 30.77 | **30.94** |
| NarrativeQA | 18.61 | **18.64** | **19.54** | 17.93 | **18.43** | 17.74 |
| Qasper | 25.83 | **27.00** | **27.00** | 26.20 | 25.39 | **26.38** |
| MultiFieldQA-en | 48.06 | **48.17** | **45.80** | 45.72 | 45.59 | **46.63** |
| MultiFieldQA-zh | **33.76** | 33.68 | **34.23** | 33.37 | **34.23** | 33.65 |
| HotpotQA | 36.35 | **36.44** | 35.79 | **36.84** | 38.00 | 37.64 |
| 2WikiMultihopQA | **25.17** | 24.61 | **25.48** | 24.07 | 24.92 | **25.98** |
| MuSiQue | **21.08** | 20.41 | **20.18** | 19.56 | **20.70** | 20.22 |
| DuReader | **30.98** | 28.62 | **29.23** | 27.36 | **29.82** | 27.91 |
| GovReport | 23.38 | **23.45** | **23.96** | 23.53 | 22.35 | **22.85** |
| QMSum | **25.45** | 24.74 | 24.59 | **24.66** | 24.67 | 24.29 |
| MultiNews | **22.63** | 21.42 | **23.30** | 21.10 | 23.53 | 21.10 |
| VCSUM | **14.19** | 13.90 | **14.21** | 13.93 | 13.86 | **13.86** |
| TREC | 39.00 | **41.83** | 39.11 | **39.49** | 37.69 | **39.49** |
| TriviaQA | 16.81 | **16.91** | 16.67 | **16.73** | **17.08** | 16.91 |
| SAMSum | 26.46 | **26.59** | **26.66** | 25.16 | **26.02** | 25.61 |
| LSHT | 31.58 | **32.00** | 24.75 | **33.25** | 29.68 | 29.96 |
| PassageCount | **1.00** | **1.00** | **1.00** | **1.00** | **1.00** | **1.00** |
| PassageRetrieval-en | **81.00** | 80.50 | 74.50 | **80.50** | 80.00 | **80.50** |
| PassageRetrieval-zh | **43.73** | 43.61 | **42.85** | 42.50 | **39.39** | 38.98 |
| LCC | 48.31 | **50.08** | 51.30 | **51.45** | 49.06 | **50.69** |
| RepoBench-P | 43.21 | **47.58** | **48.73** | 48.27 | 43.84 | **48.45** |

Table 8: Performance (%) of CateKV on larger models

| Methods | Cache | N-S1 | N-S2 | N-S3 | N-MK1 | N-MK2 | N-MK3 | FWE | N-MQ | N-MV | QA-1 | QA-2 | VT | Avg. |
|---|---|---|---|---|---|---|---|---|---|---|---|---|---|---|
| *Qwen2.5-32B* | 100% | 100.00 | 87.50 | 97.92 | 70.83 | 15.63 | 7.29 | 90.28 | 87.24 | 85.16 | 51.04 | 41.67 | 85.41 | 68.33 |
| SnapKV | 41% | 100.00 | 88.54 | 51.04 | 69.79 | 12.50 | 2.08 | 88.54 | 76.82 | 76.82 | 51.04 | 41.67 | 85.83 | 62.06 |
| PyramidKV | 41% | 100.00 | 87.50 | 46.88 | 66.67 | 8.33 | 1.04 | 84.02 | 66.93 | 67.71 | 48.96 | 41.67 | 84.79 | 58.71 |
| CateKV | 41% | 100.00 | 86.46 | 95.83 | 71.88 | 14.58 | 6.25 | 89.58 | 86.88 | 86.28 | 50.00 | 43.75 | 86.67 | 68.18 |
| *Yi-34B-200K* | 100% | 100.00 | 100.00 | 100.00 | 92.71 | 70.83 | 47.92 | 86.11 | 97.14 | 92.45 | 68.75 | 47.92 | 88.05 | 82.66 |
| SnapKV | 41% | 100.00 | 97.92 | 80.21 | 90.62 | 22.92 | 17.71 | 81.25 | 91.15 | 72.14 | 67.71 | 47.92 | 86.25 | 71.32 |
| PyramidKV | 41% | 100.00 | 100.00 | 68.75 | 91.67 | 26.04 | 12.50 | 82.99 | 91.15 | 79.43 | 69.79 | 47.92 | 86.46 | 71.39 |
| CateKV | 41% | 100.00 | 100.00 | 100.00 | 92.71 | 73.96 | 47.92 | 85.12 | 97.14 | 91.15 | 67.71 | 46.88 | 87.08 | 82.47 |
| *Phi-4-14B* | 100% | 100.00 | 97.92 | 100.00 | 100.00 | 97.92 | 100.00 | 98.96 | 98.96 | 99.22 | 80.21 | 67.71 | 100.00 | 95.08 |
| SnapKV | 43% | 100.00 | 100.00 | 7.29 | 100.00 | 93.75 | 3.13 | 99.31 | 97.66 | 99.22 | 82.29 | 66.67 | 100.00 | 79.11 |
| PyramidKV | 43% | 100.00 | 100.00 | 3.13 | 100.00 | 94.79 | 5.21 | 98.96 | 98.44 | 98.44 | 80.21 | 67.71 | 100.00 | 78.91 |
| CateKV | 43% | 100.00 | 98.96 | 100.00 | 97.92 | 98.96 | 100.00 | 99.31 | 98.44 | 99.48 | 78.13 | 67.71 | 99.79 | 94.89 |

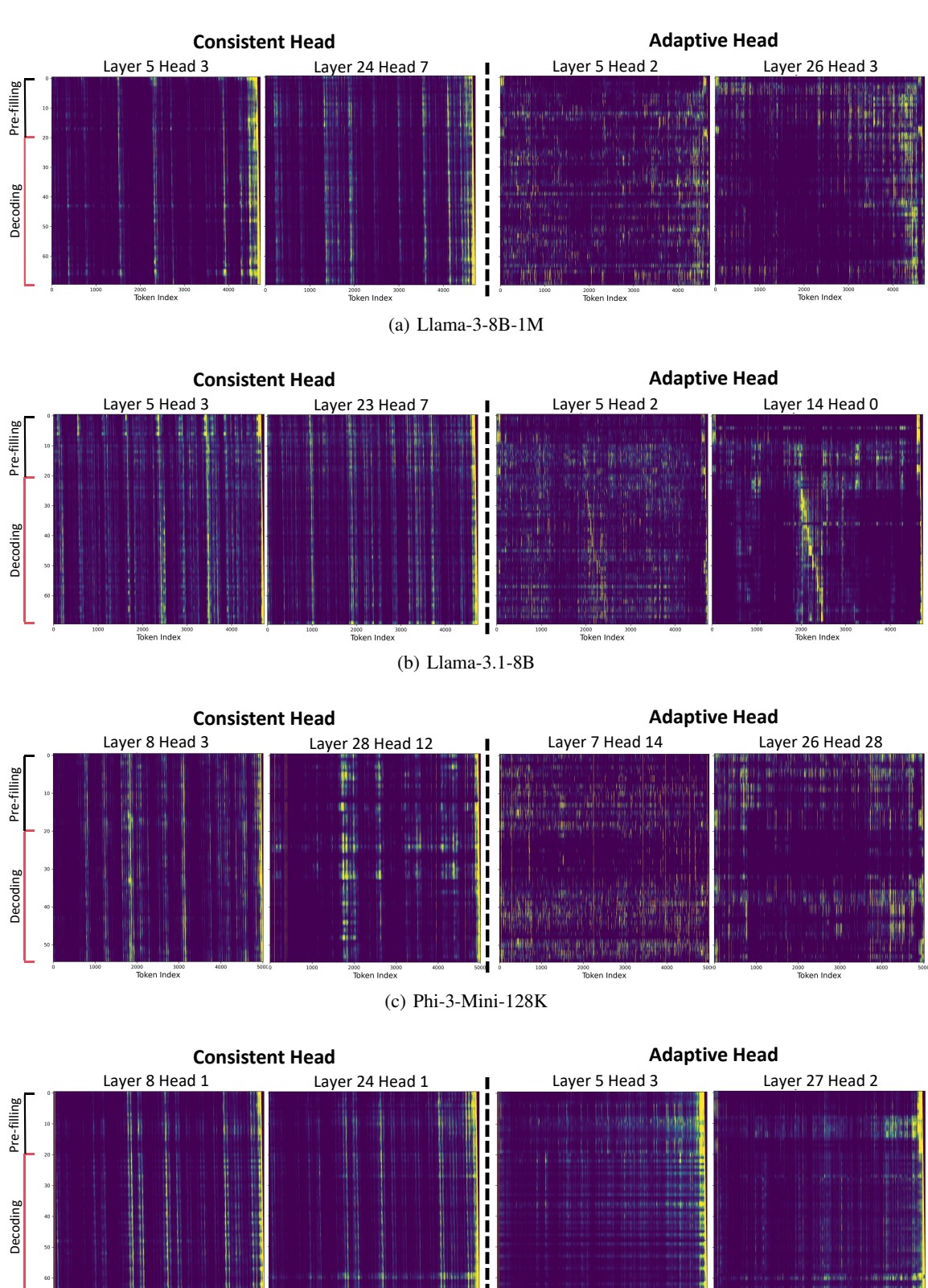

Figure 9: Sequential consistency in Llama-3-8B-1M, Llama-3.1-8B, Phi-3-Mini-128K and Yi-9B-200K

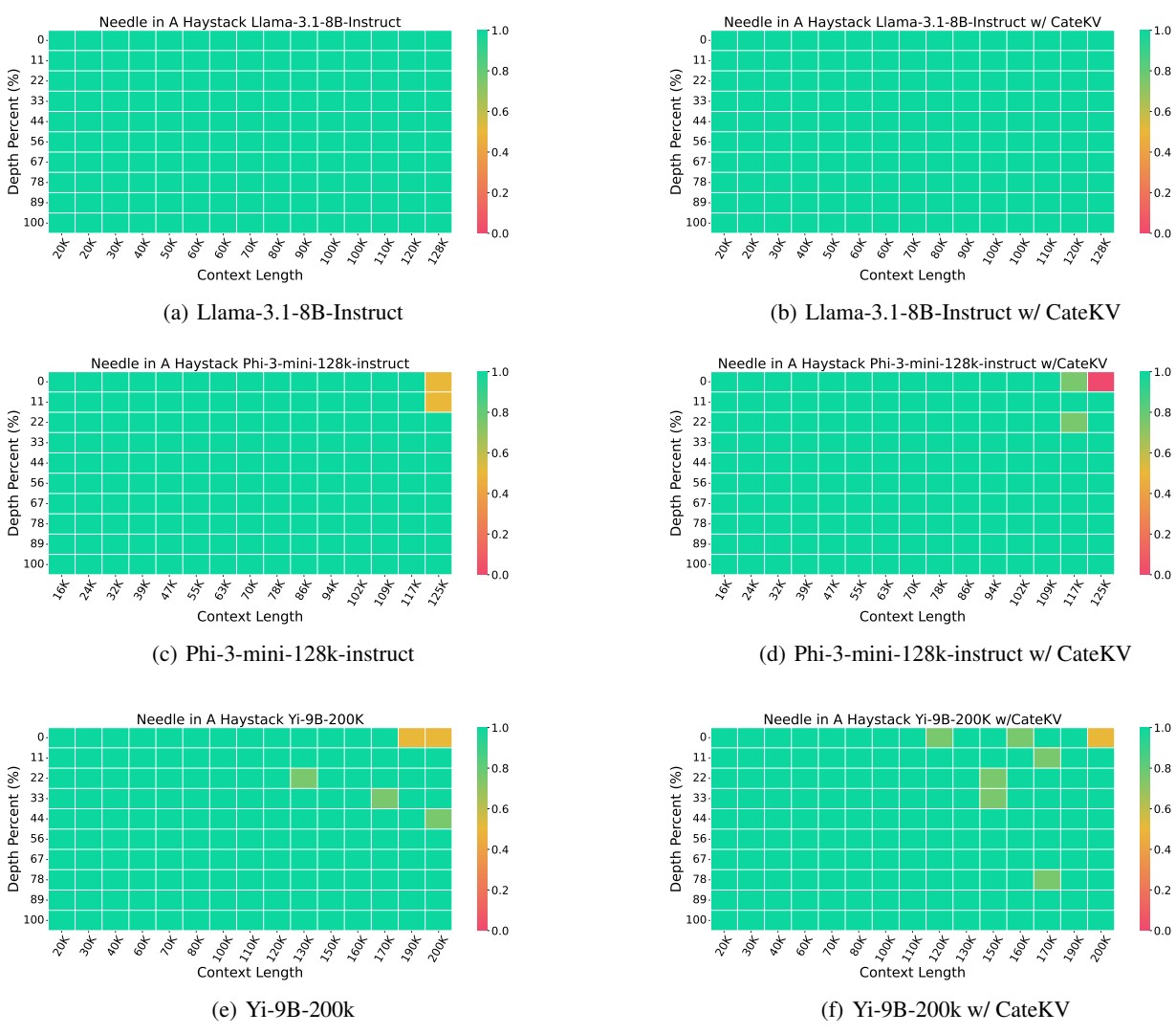

(a) Llama-3.1-8B-Instruct

(b) Llama-3.1-8B-Instruct w/ CateKV

(c) Phi-3-mini-128k-instruct

(d) Phi-3-mini-128k-instruct w/ CateKV

(e) Yi-9B-200k

(f) Yi-9B-200k w/ CateKV

Figure 10: NIAH Results on Llama-3.1-8B-Instruct(Meta AI, 2024), Phi-3-mini-128k-instruct (Abdin et al., 2024) and Yi-9B-200k (AI et al., 2024)

