# OpenReview forum: "CateKV: On Sequential Consistency for Long-Context LLM Inference Acceleration"
_ICML.cc/2025/Conference — ICML 2025 poster_

### Official Review · Reviewer_Tg7K · 2025-03-08

**Overall Recommendation:** 3

**Summary:**

This paper proposes CateKV, which improves the inference efficiency by adaptively evicting and retrieving KV cache based on sequential consistency.

**Claims And Evidence:**

Please see **Other Strengths And Weaknesses**

**Essential References Not Discussed:**

Please see **Other Strengths And Weaknesses**

**Experimental Designs Or Analyses:**

Please see **Other Strengths And Weaknesses**

**Methods And Evaluation Criteria:**

Please see **Other Strengths And Weaknesses**

**Other Comments Or Suggestions:**

Please see **Other Strengths And Weaknesses**

**Other Strengths And Weaknesses:**

**Strengths**:

1. The paper is easy to follow, with clear writing and presentation.
2. The evaluation results are good and comprehensive.

**Weaknesses**:

1. The main concern I have with this paper is the issues of scalability. In the paper, the largest model being evaluated is 9B. It would be better if the authors could provide results on larger models (30B/70B).

2. Some references [1-3] are missing in the related work for KV cache optimization.


[1] Keyformer: KV Cache Reduction through Key Tokens Selection for Efficient Generative Inference, MLSys 2024.

[2] Q-Hitter: A Better Token Oracle for Efficient LLM Inference via Sparse-Quantized KV Cache, MLSys 2024.

[3] ALISA: Accelerating Large Language Model Inference via Sparsity-Aware KV Caching, ISCA 2024.

**Questions For Authors:**

Please see **Other Strengths And Weaknesses**

**Relation To Broader Scientific Literature:**

Please see **Other Strengths And Weaknesses**

**Theoretical Claims:**

Please see **Other Strengths And Weaknesses**

---

> ### Author Rebuttal · Authors · 2025-04-01
>
> We sincerely thank the reviewer for your valuable time and constructive feedback. In the following, we provide our responses to each question.
>
> > Weaknesses 1: The main concern I have with this paper is the issues of scalability. In the paper, the largest model being evaluated is 9B. It would be better if the authors could provide results on larger models (30B/70B).
>
> We have tried our best to expand experiments on three larger models (30B&14B) in the following. Due to limited resources, we are unable to finish experiments on 70B models during this rebuttal period, but promise to include the results of 70B models in the final version.
>
> |Methods|Cache|N-S1|N-S2|N-S3|N-MK1|N-MK2|N-MK3|FWE|N-MQ|N-MV|QA-1|QA-2|VT|Avg|
> |-|-|-|-|-|-|-|-|-|-|-|-|-|-|-|
> |**Qwen2.5-32B**|100%|100.00|87.50|97.92|70.83|15.63|7.29|90.28|87.24|85.16|51.04|41.67|85.41|68.33|
> |SnapKV|41%|100.00|88.54|51.04|69.79|12.50|2.08|88.54|76.82|76.82|51.04|41.67|85.83|62.06|
> |PyramidKV|41%|100.00|87.50|46.88|66.67|8.33|1.04|84.02|66.93|67.71|48.96|41.67|84.79|58.71|
> |CateKV|41%|100.00|86.46|95.83|71.88|14.58|6.25|89.58|86.88|86.28|50.00|43.75|86.67|68.18|
> |**Yi-34B-200K**|100%|100.00|100.00|100.00|92.71|70.83|47.92|86.11|97.14|92.45|68.75|47.92|88.05|82.66|
> |SnapKV|41%|100.00|97.92|80.21|90.62|22.92|17.71|81.25|91.15|72.14|67.71|47.92|86.25|71.32|
> |PyramidKV|41%|100.00|100.00|68.75|91.67|26.04|12.50|82.99|91.15|79.43|69.79|47.92|86.46|71.39|
> |CateKV|41%|100.00|100.00|100.00|92.71|73.96|47.92|85.12|97.14|91.15|67.71|46.88|87.08|82.47|
> |**Phi-4-14B**|100%|100.00|97.92|100.00|100.00|97.92|100.00|98.96|98.96|99.22|80.21|67.71|100.00|95.08|
> |SnapKV|43%|100.00|100.00|7.29|100.00|93.75|3.13|99.31|97.66|99.22|82.29|66.67|100.00|79.11|
> |PyramidKV|43%|100.00|100.00|3.13|100.00|94.79|5.21|98.96|98.44|98.44|80.21|67.71|100.00|78.91|
> |CateKV|43%|100.00|98.96|100.00|97.92|98.96|100.00|99.31|98.44|99.48|78.13|67.71|99.79|94.89|
>
> The context length was set based on the maximum length supported by the model, with 128k for Qwen2.5-32B, Yi-34B-200K, and 16K for Phi-4-14B. The results in the above table demonstrate that CateKV scales effectively, delivering near-full-attention accuracy for 30B and 14B models, surpassing baseline methods such as SnapKV and PyramidKV.
>
> > Weaknesses 2: Some references [1-3] are missing in the related work for KV cache optimization.
>
> Thank you very much for recommending these references, and we will carefully incorporate the discussions of these papers in §2.1 (KV Cache Eviction Algorithm), which is initially summarized as follows:
>
> Keyformer proposes a score-based KV cache eviction algorithm that selectively retains only 'key' tokens with high attention weights, reducing cache size and memory bandwidth. Q-Hitter introduces a hybrid KV cache eviction criterion combining token importance (Heavy Hitters) and quantization-friendliness, enabling aggressive sparsification and low-bit quantization. ALISA designs a token-prioritization-based KV cache eviction algorithm using Sparse Window Attention (SWA) to dynamically reduce memory footprint, coupled with system-level optimizations for efficient caching-recomputation trade-offs. While these KV cache eviction methods demonstrate computational efficiency, they inherently incur information loss due to the reduction of attention across all heads or the quantization of data precision. In contrast, CateKV preserves the complete KV cache for adaptive heads, maintaining critical contextual information while still achieving efficiency through selective eviction in consistent heads.
>
> [1] Keyformer: KV Cache Reduction through Key Tokens Selection for Efficient Generative Inference, MLSys 2024.
>
> [2] Q-Hitter: A Better Token Oracle for Efficient LLM Inference via Sparse-Quantized KV Cache, MLSys 2024.
>
> [3] ALISA: Accelerating Large Language Model Inference via Sparsity-Aware KV Caching, ISCA 2024.

---

> > ### Comment · Reviewer_Tg7K · 2025-04-02
> >
> > Thank you for your response. I will increase my score from 2 to 3.

---

> > > ### Author Response · Authors · 2025-04-03
> > >
> > > Thank you very much for your support. We will carefully follow the advice to include the analysis and the references into the revision.

---

### Official Review · Reviewer_5Grq · 2025-03-13

**Overall Recommendation:** 3

**Summary:**

This paper proposes a novel KV cache algorithm that calculates the coefficient-of-variation of attention tokens at each layer. Specifically, it identifies consistent head and adaptive head where most of the KV cache can be reduced from the consistent head. In experiments, author demonstrate the effectiveness of the method on multiple long-context evaluation datasets. The performance is superior against existing KV-cache methods.

**Claims And Evidence:**

In section 3.2, the author claims "To save the computational cost, we set an observation window that contains the last query tokens of the input to identify head types and critical tokens. " while I don't think it's fully justified.

**Essential References Not Discussed:**

I am wondering why [1] is not discussed nor compared in the paper.

[1] Model tells you what to discard: Adaptive kv cache compression for llms, ICLR 23'

**Experimental Designs Or Analyses:**

Yes.

**Methods And Evaluation Criteria:**

Yes, except one point
1. Why the method cannot be applied to traiditional LLM eval dataset? What does long-context matter here?

**Other Comments Or Suggestions:**

N/A

**Other Strengths And Weaknesses:**

Strengths:
1. The paper is easy to follow with good demonstrations and examples.

**Questions For Authors:**

1. Can you apply CateKV on traditional LLM Eval tasks?

2. How is the method generalize without reference dataset ?

**Relation To Broader Scientific Literature:**

It can be used to accelerate the industry LLM serving.

**Theoretical Claims:**

This paper does not include theoretical claims.

---

> ### Author Rebuttal · Authors · 2025-04-01
>
> We sincerely thank the reviewer for your valuable time and constructive feedback. In the following, we provide our responses point-by-point.
>
> > Claims: About the design of observation window
>
> Here, we provide a more detailed explanation regarding the design of the observation window:
>
> 1. **The observation window helps avoid quadratic memory and computational costs**. In the pre-filling stage, the full attention matrix has size $n \times n$ (where $n$ is the context length). Analyzing the full attention matrix directly increases the computational complexity from $O(n)$ to $O(n^2)$, posing challenges in memory and computation, especially for long inputs. Using the full attention matrix for head classification led to an out-of-memory (OOM) error on a single A100 GPU for inputs larger than 8k tokens
>
> 2. **The attention within the observation window is more representative**. Due to the causal mask, the last query tokens capture global information similar to what is needed during decoding. Including intermediate query tokens may cause erroneous guidance, as these tokens have partial information. The table below shows that increasing the observation window size from 64 to 4k with $r = 0.4$ results in negative performance gains.
>
> |Methods|N-S1|N-S2|N-S3|N-MK1|N-MK2|N-MK3|FWE|N-MQ|N-MV|QA-1|QA-2|VT|Avg|
> |-|-|-|-|-|-|-|-|-|-|-|-|-|-|
> |CateKV ($L_{obs}=64$)|100.00|100.00|100.00|98.96|97.92|41.67|71.88|98.44|96.88|73.96|50.00|85.63|84.61|
> |CateKV ($L_{obs}=4000$)|100.00|100.00|98.96|97.92|97.92|19.79|66.67|99.21|97.92|70.83|48.96|76.25|81.20|
>
> We will enrich our explanation and add the above analysis in the submission to well justify our choice. Thank you very much for pointing out the insufficiency of this claim.
>
> > Q1: Applicability of CateKV to traditional LLM Eval tasks and the role of long-context
>
> We follow the research line of LLMs under long-context inference, which causes a significant bottleneck in memory consumption and computation latency. Regarding its practical value, some examples, like the RAG (retrieval-augmented generation) applications, long-document understanding, multi-turn interaction, etc., easily make the LLMs face long-context inference scenarios. In this case, it is critical to accelerate LLM inference to avoid the vanilla catastrophic response speed (it can be minute-level for A100 with 80G memory under 100k-long context).
>
> Naturally, the existing long-context acceleration methods, including CateKV, are applicable to traditional short-context LLM evaluation datasets. We tested the performance of CateKV and baseline methods on MMLU and MMLU-Pro, with all methods retaining approximately 25% of the KV cache and utilizing a one-shot prompting approach. The results in the table below show that CateKV still maintains performance comparable to full attention. It is worth noting that in short-context scenarios, attention to recent tokens is more likely to retain complete contextual information, allowing simpler methods (such as StreamingLLM) to maintain much of the performance.
>
> |Methods|MMLU|MMLU-Pro|
> |-|-|-|
> |Llama3-8B|59.70|30.71|
> |StreamingLLM|58.98|30.04|
> |SnapKV|59.60|30.43|
> |PyramidKV|59.60|30.43|
> |CateKV|59.60|30.71|
>
> > Essential Reference:
> [1] Model tells you what to discard: Adaptive kv cache compression for llms
>
> Thank you for the recommendation and we will incorporate the discussion of FastGen [1] in Section 2.2. For the comparison, we need to clarify that in our practice, FastGen easily incurs OOM when going beyond 8K. This is mainly because of the usage of the complete attention matrix during its pre-filling stage. We will add more explanations in the updated version.
>
> > Q2: How is the method generalize without reference dataset ?
>
> While CateKV uses a reference dataset to achieve static identification, the method proposed in Section 3.2 actually can perform without a reference dataset, yielding the dynamic identification. In the following table, we show that the dynamic identification method exhibits similar performance as the reference-based static identification method, both comparable to or surpassing full attention, highlighting the generalizability of CateKV.
>
> | Method | RULER-128K(Llama3) | Longbench(Llama3) | RULER-128K(Llama-3.1) | Longbench(Llama-3.1) |
> |-|-|-|-|-|
> | Full | 84.10 | 31.27 | 84.55 | 33.68 |
> | Dynamic ($r=0.4$) | 84.17 | 31.14 | 84.63 | 33.43 |
> | Static ($r=0.4$)| 84.61 | 31.48 | 84.66 | 33.70 |
> | Dynamic ($r=0.3$) | 83.49 | 30.61 | 81.59 | 32.89 |
> | Static ($r=0.3$) | 83.84 | 31.08 | 82.14 | 33.38 |
>
> However, the following drawbacks arise without a reference dataset:
>
> - At the pre-filling stage, the extra sample-wise computation needed for head identification increases computational overhead.
> - The need for a complete observation matrix in the pre-filling stage to assist classification prevents integration with sparsification-based pre-filling acceleration methods.
>
> We will include the above analysis in the submission to improve the clarity.

---

### Official Review · Reviewer_cc5K · 2025-03-13

**Overall Recommendation:** 4

**Summary:**

This paper studies the long-context inference acceleration of LLMs through sequential consistency patterns. By observation of distinguishing activation of attention heads, the authors classified them into consistent heads and adaptive heads, which can be used to promote the decoding acceleration. Different from previous methods that considered the acceleration of prefilling stages or decoding stages independently, the proposed method discovers the sequential consistency estimated at the prefilling stages efficiently help the decoding both in memory and computation speedup. Extensive experiments show the effectiveness of the proposed method.

**Claims And Evidence:**

The claims presented in the submission are well-supported by both intuitive understanding and empirical evidence. The authors provided the empirical statistics regarding the sequential consistency in the main part and appendix. The results demonstrated the substantial effectiveness of the proposed methods.

**Essential References Not Discussed:**

No.

**Experimental Designs Or Analyses:**

I have checked experimental parts and their analysis including the complementary results in the appendix. The experimental results are sound and adequately demonstrate the effectiveness of the proposed CateKV. The results demonstrate the consistent acceleration benefit and the gain when combined with the conventional acceleration methods.

**Methods And Evaluation Criteria:**

The authors designed a proper method to distinguish consistent heads and adaptive heads, and on that basis, the efficiency in KV cache reduction and compute speedup has been gained. The evaluation follows the common bench, which covers different dimensions including the context length, different backbones and performance against the acceleration.

**Other Comments Or Suggestions:**

- It is better to add more discussion about why not choose a soft attention weight matrix to design the method. Intuitively, this can be more precise and flexible. The authors could add more discussion in terms of this point.

**Other Strengths And Weaknesses:**

Strengths
- The proposed method captures the essential patterns during the inference of LLMs, where the evidence of the sparse computation inherent in the transformers can be estimated along with the sequential consistency. The design is natural and simple yet effective.
- The experiments and evaluations are comprehensive, covering a broad range of LLM inference benchmarks. The experimental results demonstrate the effectiveness of CateKV. Besides, the experiments combined with the previous acceleration methods demonstrate the orthogonality of CateKV.
- The paper is well-written, clearly structured, and easy to follow.

Weakness:
- It is unclear how to set the threshold to distinguish consistent heads and adaptive heads. Do we need some searching-based strategy to find the best threshold or some other ways?
- For figure 4, it seems that CateKV has the similar trend with Quest, which both are not good than Top-k. What does this mean? Is it means that the Top-k strategy is actually a good way to find the sparse compute structure? Please claim this details.
- In Figure 6, why DuoAttention has a so rapid drop in performance at the ratio of 0.4. Please provided more analysis and claimed that why CateKV could maintain the effectiveness.

**Questions For Authors:**

- What is the difference between CateKV and DuoAttention? Can you provide more qualitative comparison between the selected tokens of CateKV and those of DuoAttention. Such an result can be more convincing to reflect the novelty of CateKV.

**Relation To Broader Scientific Literature:**

LLM inference acceleration is tightly related to many areas in the recent AI communities, since it is a trend to integrate LLM to enhance the performance of many tasks. Since it is always time-intensive when the input or the output of LLMs is very long, it is very demanding to explore the inference acceleration of LLMs, which is critical to many AI applications in the era of foundation models.

**Theoretical Claims:**

Not Applicable.

---

> ### Author Rebuttal · Authors · 2025-04-01
>
> We appreciate the supportive comments and the comprehensive evaluation of our method. In the following, we provide our responses point-by-point.
>
> > Weakness 1: About setting the best threshold
>
> We select the threshold based on a fixed ratio of consistent-to-adaptive heads, as while CV scores vary significantly across inputs, the proportion of heads exhibiting consistent patterns remains remarkably stable within the model architecture.
>
> In task-agnostic scenarios, determining the optimal threshold or adaptive head ratio for each sample is challenging. One potential approach is to incorporate attention weight recall as an additional criterion in the head classification process. If the attention weight recall exceeds a threshold (denoted as $t$), these heads can also adopt the consistent pattern.
>
> As shown in the table below (LLaMA-3-8B,$r=0.4$,$t=0.98$), this recall-threshold method enables input-adaptive ratio adjustment while maintaining performance (Avg 84.32% vs 84.61% CateKV). However, the marginal gains suggest this approach may not be universally optimal. We will further explore dynamic threshold adjustment methods in future work.
>
> |Methods|N-S1|N-S2|N-S3|N-MK1|N-MK2|N-MK3|FWE|N-MQ|N-MV|QA-1|QA-2|VT|Avg|
> |-|-|-|-|-|-|-|-|-|-|-|-|-|-|
> |CateKV ($r=0.4$)|100.00|100.00|100.00|98.96|97.92|41.67|71.88|98.44|96.88|73.96|50.00|85.63|84.61|
> |CateKV+recall threshold|100.00|100.00|100.00|98.96|97.92|42.71|71.88|98.18|96.35|71.87|50.00|83.96|84.32|
> |Average adaptive head ratio|34.38%|37.13%|37.16%|37.13%|38.67%|36.72%|35.54%|37.23%|37.20%|37.25%|37.89%|32.03%|36.53%|
>
> For task-aware scenarios, please refer to our response to Reviewer 3ByN'Weakness 2, where we discuss dynamic ratio selection in task-aware contexts.
>
> > Weakness 2: About the meaning of Topk curve.
>
> Figure 4 compares different methods' ability to approximate the ground-truth top-k attention weights during decoding. The top-k curve represents the upper bound of achievable recall, as it directly uses the actual highest attention weights (which are unknown in advance during real inference).
>
> This does not imply top-k is universally "better," but rather that it serves as the theoretical optimum. The key advantage of CateKV/Quest is that they avoid computing full attention, making them practical for real-world deployment.
>
> >Weakness 3 and Question: Comparison Between CateKV and DuoAttention
>
> 1. Differences between CateKV and DuoAttention: From a conceptual perspective, consistent heads emphasize the overall similarity of attention across different sequences, while Streaming Heads prioritize whether initial and recent tokens cover most of the attention. Thus, in terms of token selection, streaming heads only focus on the initial and recent tokens for attention, whereas consistent heads have a broader search space. To further illustrate this, we randomly selected 100 samples from WikiText and performed inference using LLama3-8B. Our analysis shows that approximately 72% of the tokens selected by consistent heads do not belong to the initial or recent tokens, demonstrating greater flexibility compared to streaming heads.
>
> 2. Analysis of the rapid drop: From the perspective of attention recall, the number of streaming heads within a model is inherently limited. As shown in Figure 4, a larger proportion of heads recall less than 0.8 of the attention weights under the streaming pattern. An overreliance on the streaming pattern results in significant information loss, which explains the rapid drop in the performance of DuoAttention when the ratio is below 0.4. In contrast, even with an aggressive increase in the proportion of consistent heads, the consistent pattern effectively preserves critical tokens, including those that capture global information from prior attention. These retained tokens enhance the robustness of CateKV, making it more effective than DuoAttention.
>
> > Suggestion: Using Soft Attention Weight Matrix for Method Design
>
> Utilizing a soft attention weight matrix is an intuitive approach. However, we believe that the values in the soft attention weight matrix reflect the importance of individual tokens, whereas the binarized observation matrix used in CateKV more effectively captures the relative importance of tokens. Since sparse attention methods aim to identify the top-k important positions, sequence similarity should be evaluated by the alignment of their top-k attention indices, which reflects a relative relationship. Additionally, the values of attention weights fluctuate with the query, and attention sinks in some heads complicate the use of a soft attention matrix for classification.
>
> We also conducted an experiment where we removed the binarization step from the identification process (a soft version). When $r = 0.5$, on the RULER-128K niah-multikey3 task, the performance dropped by 14.58 points (from 63.54 to 48.96) compared to full attention on LLama-3.1-8B.

---

> > ### Comment · Reviewer_cc5K · 2025-04-07
> >
> > Thank you for the authors' substantial effort in addressing my concerns. Overall, I think the idea is novel that utilizes the prefilling knowedge to promote decoding speedup. After considering the authors' rebuttal to my review and other reviewers' questions, I tend to maintain the acceptance of this submission and hope the authors carefully incorporate the suggestions.

---

> > > ### Author Response · Authors · 2025-04-08
> > >
> > > Thank you very much for your recognition and support of our work. We will carefully consider the reviewers' suggestions and incorporate them in the subsequent version.

---

### Official Review · Reviewer_3ByN · 2025-03-14

**Overall Recommendation:** 3

**Summary:**

CateKV introduces a hybrid KV cache optimization method that improves long-context LLM inference efficiency by leveraging sequential consistency in attention heads. The key insight is that certain attention heads exhibit stable attention patterns across both pre-filling and decoding stages, while others remain highly dynamic. Based on this observation, CateKV: (1) Classifies attention heads into two types:
Consistent heads: Retain only a subset of KV pairs based on pre-filling attention patterns. Adaptive heads: Retain most KV pairs to ensure flexible attention computation. (2) Uses a coefficient-of-variation (CV)-based scoring algorithm to differentiate between consistent and adaptive heads. (3) Applies selective KV retention, where consistent heads store a minimal set of critical tokens, while adaptive heads retain most of their KV pairs.

**Claims And Evidence:**

Yes.

**Essential References Not Discussed:**

[1] MagicPIG: LSH Sampling for Efficient LLM Generation

**Experimental Designs Or Analyses:**

The experiments are well-structured but could be strengthened with:
- Latency breakdown.

**Methods And Evaluation Criteria:**

Yes.

**Other Comments Or Suggestions:**

See Strengths And Weaknesses

**Other Strengths And Weaknesses:**

Strengths
- Novel Sequential Consistency-Based KV Management. Unlike prior work that heuristically prunes KV caches (SnapKV, StreamingLLM) or dynamically retrieves KV pairs (Quest, ShadowKV), CateKV exploits consistent attention patterns across layers.

Weaknesses
- Assumption That Pre-Filling Attention Patterns Remain Consistent During Decoding. CateKV relies on the assumption that stable attention patterns persist, but certain adaptive behaviors in retrieval tasks (e.g., dynamically changing focus) may lead to inconsistencies.
- Fixed Ratio of Adaptive vs. Consistent Heads. The paper fixes the adaptive head ratio (r = 0.4) without dynamically adjusting it based on task or context length, which may be suboptimal in some settings.
- No Explicit Theoretical Bound on Eviction Performance. The CV-based classification method is empirically justified but lacks a formal guarantee on its worst-case token retention accuracy.

**Questions For Authors:**

Thank you to the authors for their latest response. Although the underlying assumptions do not fully convince me, I will raise my score to a 3, considering that similar work has been accepted at top conferences.

**Relation To Broader Scientific Literature:**

- Head-Level KV Compression Methods
- Hybrid KV Retention Approaches

**Theoretical Claims:**

No theory in this paper.

---

> ### Author Rebuttal · Authors · 2025-04-01
>
> We sincerely thank the reviewer for your valuable time and constructive feedback. Below, we provide our responses point-by-point.
>
> > Experiments: Latency breakdown
>
> We very appreciate the reviewer's constructive suggestion, and follow the advice to present a fine-grained latency analysis ([here](https://anonymous.4open.science/r/CateKV-rebuttal-1A3F/latency_breakdown.jpg)) for your reference.
>
> >Missing References
>
> Thank you for your recommendation. We will carefully include MagicPIG and references suggested by other reviewers in the discussions of the submission.
>
> >W1: Assumption that ... may lead to inconsistencies.
>
> Given a certain long-context input, we conjecture that the functionality of each head can be well characterized, even for the head where the focus may change. Informally, it resembles the functionality of human brain regions, which are categorized to play different roles. This pattern stability is well supported by empirical findings in DuoAttention, our cross-model consistency ([here](https://anonymous.4open.science/r/CateKV-rebuttal-1A3F/cross_model.jpg)) and cross-task stability analysis ([here](https://anonymous.4open.science/r/CateKV-rebuttal-1A3F/cross_task.jpg)), which we believe is inherently related to the working mechanism of LLMs. Regarding the implication of this phenomenon, we would like to give a plausible explanation: during inference, LLMs rely on both
>
> - Heads that dynamically adapt to task-specific information, i.e., adaptive heads. For retrieval tasks with dynamic focus shifts, adaptive heads retaining more KV cache specifically handle such varying information.
>
> - Heads that consistently attend to globally important tokens, i.e., consistent heads. Figure in this [link](https://anonymous.4open.science/r/CateKV-rebuttal-1A3F/Similar_focus.jpg) demonstrates that the positions focused on by consistent heads exhibit strong similarity, indicating that some tokens are crucial for the model's understanding globally.
>
> >W2: Fixed Ratio of Adaptive vs. Consistent Heads.
>
> First, the reason that we chose a fixed ratio here is that we focus on the task-agnostic scenario where previous works primarily considered. This ensures a fair comparison with the baselines under the same settings.
>
> Second, CateKV can easily adapt to task-aware scenarios by setting task-adaptive head ratios. The table in this [link](https://anonymous.4open.science/r/CateKV-rebuttal-1A3F/task_aware.jpg) exhibits different adaptive head ratios for tasks on RULER-128K (LLaMA-3.1-8B), where adjusting the ratio per task allows for lower ratios without sacrificing accuracy.
>
> We will add a more detailed discussion in the revised submission.
>
> >W3: No Explicit Theoretical ... retention accuracy.
>
> To address the concern, we present the following theoretical analysis of CateKV.
>
> **Lemma 1:** Let $G$ denote our CV-based function hypothesis, $F$ denote the real-value class defined by the binary cross-entropy loss composite on $G$, and $N$ denote the sample number of the reference dataset. Then, with the probability $\delta$, we have the Rademacher complexity bound,
>
> $\forall f\in F,  P_{\text{head}}\left(\mathbb{E}[f]-\frac{1}{N}\sum_{n=1}^Nf_n \leq 2\mathcal{R}_N(F)+\sqrt{\frac{2\log{\frac{2}{\delta}}}{N}}\right)\geq 1-\delta$,
>
> where $\mathcal{R}_N(F)$ is the conditional Rademacher average.
>
> Let $P_1$, $P_2$ denote respectively the probabilities of correctly classified consistent heads and adaptive heads, and $\bar{P}\_1$, $\bar{P}\_2$ denote respectively the probabilities of misclassified consistent heads and adaptive heads, where we have $P\_1+P\_2=P\_{\text{head}}$ and $\bar{P}\_1+\bar{P}\_2=1-P\_{\text{head}}$. Then, we can decompose the probability in the above lemma with fine-grained analysis as follows.
>
> **Theorem 1:** Let $\eta_1$, $\eta_2$ denote respectively the retention ratios of consistent heads and adaptive heads, and $\eta_1^*$, $\eta_2^*$ denote their optimal retention ratios correspondingly. Define the retion accuracy of different cases $r_{i,j}=\eta_i^*\mathbb{1}[\eta_j > \eta_i^*]+\eta_j(1-\mathbb{1}[\eta_j > \eta_i^*])$ by comparsing the retention budgets with the optimal budgets, and the hypothesis that the query attention score is the best description in order holds with the probability $\lambda$. Then, the token retention accuracy has the following inequalities,
>
> $P\_{\text{token}} = \lambda(r_{1,1}P\_1 + r\_{2,2}P\_2 + r\_{2,1}\bar{P}\_1 + r\_{1,2}\bar{P}\_2)\geq \lambda \left(\min(r\_{2,1}, r\_{1,2}) + \left[\min(r\_{1,1}, r\_{2,2})-\min(r\_{2,1}, r\_{1,2})\right]P\_{\text{head}}\right)$
>
> In this theorem, three factors, i.e., $\lambda$, budge control part and head identification accuracy $P_{\text{head}}$ make critical effect about the worst-tken retention accuracy in CateKV, which are actually also verified in the submission (like Figures 3, 4, 7 and Tables 2, 3). Due to space limitation, we will give more details and remarks on this theorem in the reviewer-author discussion phase.

---

> > ### Comment · Reviewer_3ByN · 2025-04-07
> >
> > I appreciate the authors' thoughtful rebuttal to my questions. While the explanation and references regarding the assumption of consistent heads—that is, heads which consistently attend to globally "important" tokens—are noted, I remain unconvinced by this assumption. Specifically, I am uncertain about the actual "importance" of these tokens to understanding or reasoning over the entire context. Are they really important, or could their prominence be attributed to other underlying factors? Furthermore, in extreme cases, might the importance of these tokens diminish as the generation length increases?
> >
> > Since LLMs are not explicitly trained on such assumptions, it is unclear whether this observation will generalize to future LLMs or if it could be mitigated during pre-training. Because if these tokens are not truly important, or may become less important in future contexts, this assumption could even be detrimental to model performance.
> >
> > Overall, I will maintain my current score, though I remain open to further discussion.

---

> > > ### Author Response · Authors · 2025-04-08
> > >
> > > We sincerely appreciate the reviewer’s valuable feedback. First of all, we would like to append some insights for the theoretical bound due to the space limitation in 1st-round rebuttal.
> > >
> > > **Remark 1:** From the Theorem 1, three critical factors matter w.r.t. the worst-token retention accuracy (i.e., the lower bound):
> > > - $\lambda$, whether we can find an effective and efficient measure to characterize the token correlation by score order that is correct in a probability as large as possible.
> > > - budget control, whether we can set the proper budget that can maximal gain by reducing most tokens when heads are correctly classified and simultaneously weakening the negative effect when heads are misclassified.
> > > - $P_{\text{head}}$, whether CV-based method can classify the head type as accurately as possible during token reduction.
> > >
> > > These three factors, i.e., $\lambda$, budge control part and head identification accuracy $P_{\text{head}}$ make critical effect about the worst-tken retention accuracy in CateKV, which are actually also verified in the submission (like Figures 3, 4, 7 and Tables 2, 3).
> > >
> > > Then, we would like to figure out **whether the reviewer's concern now remains on the retionality of the assumption**, so that we would not miss some other points. In the following, we give tri-fold discussion about the assumption.
> > >
> > > - **Similar phenomenon has been observed in prior works and leveraged (but different from CateKV).** Specifically, in *H2O* [1], the authors discovered that a small number of influential tokens, called heavy-hitters (H2), are sustainably crucial during generation, and in the Observation Section of *SnapKV* [2], the authors mention that consistent patterns can be identified prior to generation and remain consistent throughout the process, though these works have not found the head discrepancy in this phenomenon. In contrast, *DuoAttention* [3] found some head discrepancy but has not considered the token consistency. Prior works implicitly support the retionality of our assumption.
> > >
> > > - **The prominence of these tokens has several underlying factors, but still matters.** Except intrinsic importance of some tokens as supporting facts for generation, other tokens like initial tokens, punctuation marks, delimiters, spaces, and tokens positioned at the beginning or end of a sequence [4] also induce the consistent prominence, namely, attention sink. While these tokens may not necessarily hold intrinsic meaning, they are vital for the model to comprehend the overall context [4][5]. In the other words, even though some tokens have void semantic meaning, their functionality to relieve attention on other semantically meaningful tokens matters for LLM inference.
> > >
> > > - **Biologically plausible generalization of the consistent head assumption for future LLMs.** In long-context scenarios, the necessity of these globally important tokens becomes evident. Similar to the human brain, which selectively focuses on key pieces of information (such as chapter titles or conclusions in a narrative) to recall and maintain context, LLMs focus on crucial tokens that allow them to better understand and recall contextual information. This selective attention mechanism helps the model efficiently process extensive inputs and maintain performance in long-context scenarios. Thus, we believe that the head assumption observed in such contexts is like some sparsity entangled with LLM pre-training, which is reasonable, necessary and generalizble to future LLMs.
> > >
> > > In conclusion, we maintain that the assumption of consistent attention to certain tokens and heads, especially in long-context tasks, is a valid and beneficial strategy. This assumption is supported by both previous studies and our empirical observations. But we do appreciate the reviewer about his/her comments, even though the reviewer challenged our assumption. It is positive for research as the normal debate, and we would like to hear further comments about our above points.
> > >
> > > PS. If the reviewer has further comments but cannot add follow-up session, directly editing the previous session is also possible and we will keep reading if there are some update and interact with the reviewer. Thank you very much.
> > >
> > > [1] H2O: Heavy-Hitter Oracle for Efficient Generative Inference of Large Language Models
> > >
> > > [2] SnapKV: LLM Knows What You are Looking for Before Generation
> > >
> > > [3] DuoAttention: Efficient Long-Context LLM Inference with Retrieval and Streaming Heads
> > >
> > > [4] Active-Dormant Attention Heads: Mechanistically Demystifying Extreme-Token Phenomena in LLMs
> > >
> > > [5] Efficient Streaming Language Models with Attention Sinks

---

### Decision · Program_Chairs · 2025-05-01

**Decision:**

Accept (poster)

**Comment:**

The paper introduces CateKV, a hybrid KV cache optimization method designed to enhance the efficiency of inference in large language models (LLMs) processing long-context tasks. CateKV leverages the insight that certain attention heads demonstrate stable attention patterns, allowing selective retention of key-value (KV) pairs to significantly reduce memory and computational overhead. The reviewers agree that the method presented is novel and well-motivated, distinguishing itself from prior approaches by systematically exploiting sequential consistency in attention heads.

However, reviewers identified several concerns that authors have mostly addressed in the rebuttal:
+ The reliance on the assumption of persistent attention patterns during decoding potentially limits the method’s applicability in tasks with dynamic retrieval behaviors. The authors provided additional analyses demonstrating cross-model and cross-task stability, partially mitigating this concern.
+ The fixed ratio of adaptive vs. consistent heads could be suboptimal across various tasks and contexts. The authors provided task-aware analyses to suggest potential adaptations for different scenarios, though dynamic ratio adjustments were not originally explored.
+ A theoretical analysis and bound on retention accuracy were initially lacking. In response, authors presented a theoretical Rademacher complexity bound to justify their empirical findings.

The authors did a very good job addressing reviewers' concerns, and reviewers were also quite responsible during the rebuttal period. Overall, the work is considered a solid contribution to ICML, given its practical significance and promising empirical validation. Therefore, I'm recommending acceptance of this work.